# Design from Policies: Conservative Test-Time Adaptation for Offline Policy Optimization

**Jinxin Liu**[1,2]    **Hongyin Zhang**[1,2]    **Zifeng Zhuang**[1,2]    **Yachen Kang**[1,2]

**Donglin Wang**[1][*]    **Bin Wang**[3]

[1]Westlake University    [2]Zhejiang University    [3]Huawei Noah's Ark Lab

## Abstract

In this work, we decouple the iterative bi-level offline RL (value estimation and policy extraction) from the offline training phase, forming a non-iterative bi-level paradigm and avoiding the iterative error propagation over two levels. Specifically, this non-iterative paradigm allows us to conduct inner-level optimization (value estimation) in training, while performing outer-level optimization (policy extraction) in testing. Naturally, such a paradigm raises three core questions that are *not* fully answered by prior non-iterative offline RL counterparts like reward-conditioned policy: Q1) What information should we transfer from the inner-level to the outer-level? Q2) What should we pay attention to when exploiting the transferred information for safe/confident outer-level optimization? Q3) What are the benefits of concurrently conducting outer-level optimization during testing? Motivated by model-based optimization (MBO), we propose DROP (**D**esign f**RO**m **P**olicies), which fully answers the above questions. Specifically, in the inner-level, DROP decomposes offline data into multiple subsets and learns an MBO score model (A1). To keep safe exploitation to the score model in the outer-level, we explicitly learn a behavior embedding and introduce a conservative regularization (A2). During testing, we show that DROP permits test-time adaptation, enabling an adaptive inference across states (A3). Empirically, we find that DROP, compared to prior non-iterative offline RL counterparts, gains an average improvement probability of more than 80%, and achieves comparable or better performance compared to prior iterative baselines.

## 1 Introduction

Offline reinforcement learning (RL) [37, 40] describes a task of learning a policy from static offline data. Due to the overestimation of values at out-of-distribution (OOD) state-actions, recent iterative offline RL methods introduce various policy/value regularization to avoid deviating from the offline data distribution in the training phase. Then, these methods directly deploy the learned policy in an online environment to test its performance. To unfold our following analysis, we term this kind of learning procedure as *iterative bi-level offline RL* (Figure 1 *left*), where the inner-level optimization refers to trying to eliminate the OOD issue, the outer-level optimization refers to trying to infer an optimal policy that will be employed at testing. Here, we use the "iterative" term to emphasize that the inner-level and outer-level are iteratively optimized in the training phase. However, due to the iterative error exploitation and propagation [6] over the two levels, performing such an iterative bi-level optimization completely in training often struggles to learn a stable policy/value function.

---

[*]Corresponding author: Donglin Wang <wangdonglin@westlake.edu.cn>

37th Conference on Neural Information Processing Systems (NeurIPS 2023).

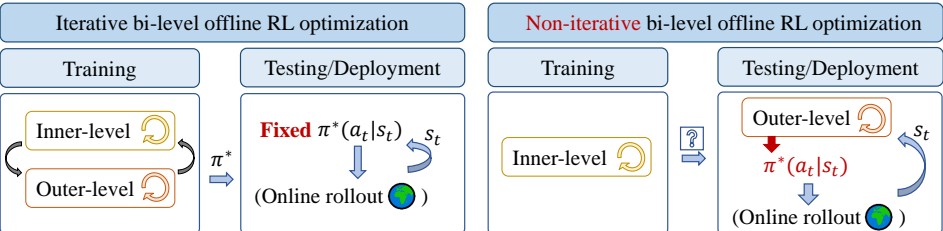

Figure 1: **Framework of (*iterative* and *non-iterative*) bi-level offline RL optimization**, where the inner-level optimization refers to regularizing the policy/value function (for OOD issues) and the outer-level refers to updating the policy for reward maximizing. The "⍰" transferred from inner-level to outer-level depends on the specific choice of algorithm used (see different choices for ⍰ in Table 1).

In this work, we thus advocate for *non-iterative bi-level optimization* (Figure 1 *right*) that decouples the bi-level optimization from the training phase, *i.e.*, performing inner-level optimization (eliminating OOD) in training and performing outer-level optimization (maximizing return) in testing. Intuitively, incorporating the outer-level optimization into the testing phase can eliminate the iterative error propagation over the two levels, *i.e.*, there is no iteratively learned value function that bootstraps off of itself and thus propagates errors further. Then, three core questions are: Q1) What information ("⍰") should we transfer from the inner-level to the outer-level? Q2) What should we pay special attention to when exploiting "⍰" for safe/confident outer-level optimization? Q3) Notice that the outer-level optimization and the testing rollout form a new loop, what new benefit does this give us?

Intriguingly, prior works under such a non-iterative framework have proposed to transfer ("⍰" in Q1) filtered trajectories [10], a reward-conditioned policy [14, 34], and the Q-value estimation of the behavior policy [6, 24], all of which, however, partially address the above questions (we will discuss these works in Section 3.4). In this work, we thus propose a new alternative method that transfers an embedding-conditioned score model (Q-value) and we will show that this method sufficiently answers the raised questions and benefits most from the non-iterative bi-level framework.

Before introducing our method, we introduce a conceptually similar task (to the non-iterative bi-level optimization) — offline model-based optimization (MBO[2] [62]), which aims to discover, from static input-score pairs, a new design input that will lead to the highest score. Typically, offline MBO first learns a score model that maps the input to its score via supervised regression (*corresponding to inner-level optimization*), and then conducts inference with the learned score model (as "⍰"), for instance, by optimizing the input against the learned score model via gradient ascent (*corresponding to the outer-level*). To enable this MBO implementation in offline RL, we are required to decompose an offline RL task into multiple *sub-tasks*, each of which thus corresponds to a behavior policy-return (parameters-return) pair. However, there are practical optimization difficulties when using high-dimensional policy's parameter space (as input for the score model) to learn such a score model (*inner-level*) and conduct inference (*outer-level*). At inference, directly extrapolating the learned score model ("⍰") also tends to drive the high-dimensional candidate policy parameters towards OOD, invalid, and low-scoring parameters [32], as these are falsely and over-optimistically scored by the learned score model in inner-level optimization.

To tackle these problems, we suggest[3] A1) learning low-dimensional embeddings for those sub-tasks decomposed in the MBO implementation, over which we estimate an embedding-conditioned Q-value ("⍰" in Q1), and A2) introducing a conservative regularization, which pushes down the predicted scores on OOD embeddings. Intuitively, this conservation aims to avoid over-optimistic exploitation and protect against producing unconfident embeddings when conducting outer-level optimization (Q2). Meanwhile, we suggest A3) conducting test-time adaptation, which means we can dynamically adjust the inferred embeddings across testing states (aka deployment adaptation). We name our method DROP (**D**esign f**RO**m **P**olicies). Compared with standard offline MBO for parameter design [62], test-time adaptation in DROP leverages the sequential structure of RL tasks, rather than simply conducting inference at the beginning of testing rollout. Empirically, we demonstrate that DROP can effectively extrapolate a better policy that benefits from the non-iterative framework by answering the raised questions, and achieves better performance compared to state-of-the-art offline RL algorithms.

---

[2]Please note that **this MBO is different from the typical model-based RL**, where the *model* in MBO denotes *a score model* while *that* in the model-based RL denotes *a transition dynamics model*.

[3]We use A1, A2, and A3 to denote our answers to the raised questions (Q1, Q2, and Q3) respectively.

## 2 Preliminaries

**Reinforcement learning.** We model the interaction between agent and environment as a Markov Decision Process (MDP) [61], denoted by the tuple $(\mathcal{S}, \mathcal{A}, P, R, p_0)$, where $\mathcal{S}$ is the state space, $\mathcal{A}$ is the action space, $P : \mathcal{S} \times \mathcal{A} \times \mathcal{S} \to [0, 1]$ is the transition kernel, $R : \mathcal{S} \times \mathcal{A} \to \mathbb{R}$ is the reward function, and $p_0 : \mathcal{S} \to [0, 1]$ is the initial state distribution. Let $\pi \in \Pi := \{\pi : \mathcal{S} \times \mathcal{A} \to [0, 1]\}$ denotes a policy. In RL, we aim to find a stationary policy that maximizes the expected discounted return $J(\pi) := \mathbb{E}_{\tau \sim \pi} \left[ \sum_{t=0}^{\infty} \gamma^t R(\mathbf{s}_t, \mathbf{a}_t) \right]$ in the environment, where $\tau = (\mathbf{s}_0, \mathbf{a}_0, r_0, \mathbf{s}_1, \mathbf{a}_1, \dots)$, $r_t = R(\mathbf{s}_t, \mathbf{a}_t)$, is a sample trajectory and $\gamma \in (0, 1)$ is the discount factor. We also define the state-action value function $Q^{\pi}(\mathbf{s}, \mathbf{a}) := \mathbb{E}_{\tau \sim \pi} \left[ \sum_{t=0}^{\infty} \gamma^t R(\mathbf{s}_t, \mathbf{a}_t) | \mathbf{s}_0 = \mathbf{s}, \mathbf{a}_0 = \mathbf{a} \right]$, which describes the expected discounted return starting from state $\mathbf{s}$ and action $\mathbf{a}$ and following $\pi$ afterwards, and the state value function $V^{\pi}(\mathbf{s}) = \mathbb{E}_{\mathbf{a} \sim \pi(\mathbf{a}|\mathbf{s})} [Q^{\pi}(\mathbf{s}, \mathbf{a})]$. To maximize $J(\pi)$, the actor-critic algorithm alternates between policy evaluation and improvement. Given initial $Q^0$ and $\pi^0$, it iterates

$$Q^{k+1} \leftarrow \arg\min_Q \ \mathbb{E}_{(\mathbf{s}, \mathbf{a}, \mathbf{s}') \sim \mathcal{D}^+, \mathbf{a}' \sim \pi^k(\mathbf{a}'|\mathbf{s}')} \left[ (R(\mathbf{s}, \mathbf{a}) + \gamma Q^k(\mathbf{s}', \mathbf{a}') - Q(\mathbf{s}, \mathbf{a}))^2 \right], \quad (1)$$

$$\pi^{k+1} \leftarrow \arg\max_\pi \ \mathbb{E}_{\mathbf{s} \sim \mathcal{D}^+, \mathbf{a} \sim \pi(\mathbf{a}|\mathbf{s})} \left[ Q^{k+1}(\mathbf{s}, \mathbf{a}) \right], \quad (2)$$

where the value function (critic) $Q(\mathbf{s}, \mathbf{a})$ is updated by minimizing the mean squared Bellman error with an experience replay $\mathcal{D}^+$ and, following the deterministic policy gradient theorem [59], the policy (actor) $\pi(\mathbf{a}|\mathbf{s})$ is updated to maximize the estimated $Q^{k+1}(\mathbf{s}, \pi(\mathbf{a}|\mathbf{s}))$.

**Offline reinforcement learning.** In offline RL [40], the agent is provided with a static data $\mathcal{D} = \{\tau\}$ which consists of trajectories collected by running some data-generating policies. Note that here we denote static offline data $\mathcal{D}$, distinguishing from the experience replay $\mathcal{D}^+$ in the online setting. Unlike the online RL problem where the experience $\mathcal{D}^+$ in Equation 1 can be dynamically updated, the agent in offline RL is not allowed to interact with the environment to collect new experience data. As a result, naively performing policy evaluation as in Equation 1 may query the estimated $Q^k(\mathbf{s}', \mathbf{a}')$ on actions that lie far outside of the static offline data $\mathcal{D}$, resulting in pathological value $Q^{k+1}(\mathbf{s}, \mathbf{a})$ that incurs large error. Further, iterating policy evaluation and improvement will cause the inferred policy $\pi^{k+1}(\mathbf{a}|\mathbf{s})$ to be biased towards OOD actions with erroneously overestimated values.

**Offline model-based optimization.** Model-based optimization (MBO) [63] aims to find an optimal design input $\mathbf{x}^*$ with a given score function $f^* : \mathcal{X} \to \mathcal{Y} \subset \mathbb{R}$, *i.e.*, $\mathbf{x}^* = \arg\max_\mathbf{x} f^*(\mathbf{x})$. Typically, we can repeatedly query the oracle score model $f^*$ for new candidate design, until it produces the best design. However, we often do not have the oracle score function $f^*$ but are provided with a static offline dataset $\{(\mathbf{x}, y)\}$ of labeled input-score pairs. To track such a *design from data* question, we can fit a parametric model $f$ to the static offline data $\{(\mathbf{x}, y)\}$ via the empirical risk minimization (ERM): $f \leftarrow \arg\min_f \mathbb{E}_{(\mathbf{x}, y)} \left[ (f(\mathbf{x}) - y)^2 \right]$. Then, starting from the best point in the dataset, we can perform gradient ascent on the design input and set the inferred optimal design $\mathbf{x}^* = \mathbf{x}_K^\circ := \text{GradAscent}_f(\mathbf{x}_0^\circ, K)$, where

$$\mathbf{x}_{k+1}^\circ \leftarrow \mathbf{x}_k^\circ + \eta \nabla_\mathbf{x} f(\mathbf{x})|_{\mathbf{x} = \mathbf{x}_k^\circ}, \text{ for } k = 0, \dots, K - 1. \quad (3)$$

For simplicity, we will omit subscript $f$ in $\text{GradAscent}_f$. Since the aim is to find a better design beyond all the designs in the dataset while directly optimizing score model $f$ with ERM can not ensure new candidates (OOD designs/inputs) receive confident scores, one crucial requirement of offline MBO is thus to conduct confident extrapolation.

## 3 DROP: Design from Policies

We present our framework in Figure 2. In Sections 3.1 and 3.2, we will answer questions Q1 and Q2, setting a learned MBO score model as "⍰" (A1) and introducing a conservative regularization over the score model (A2). In Section 3.3, we will answer Q3, where we show that conducting outer-level optimization during testing can produce an adaptive embedding inference across states (A3).

### 3.1 Task Decomposition

Our core idea is to explore MBO in the non-iterative bi-level offline RL framework (Figure 1 *right*), while capturing the sequential characteristics of RL tasks and answering the raised questions (Q1, Q2, and Q3). To begin with, we first decompose the offline data $\mathcal{D}$ into $N$ offline subsets

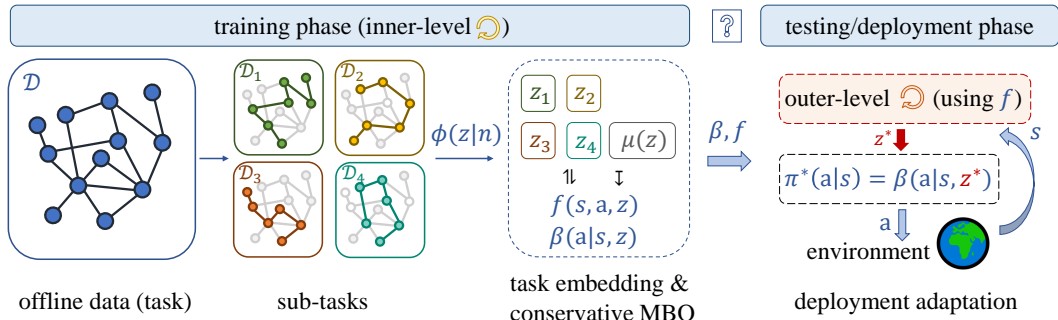

Figure 2: **Overview of DROP.** Given offline dataset $\mathcal{D}$, we decompose the data into $N$ ($= 4$ in the diagram) subsets $\mathcal{D}_{[N]}$. Over the decomposed sub-tasks, we learn a task embedding $\phi(\mathbf{z}|n)$ and conduct conservative MBO by learning a score model $f(\mathbf{s}, \mathbf{a}, \mathbf{z})$ and $N$ behavior policies (modeled by a contextual policy $\beta(\mathbf{a}|\mathbf{s}, \mathbf{z})$). During test-time, at state $\mathbf{s}$, we perform test-time adaptation by adapting the policy (contextual variable) with $\pi^*(\mathbf{a}|\mathbf{s}) = \beta(\mathbf{a}|\mathbf{s}, \mathbf{z}^*)$, where $\mathbf{z}^* = \arg\max_{\mathbf{z}} f(\mathbf{s}, \beta(\mathbf{a}|\mathbf{s}, \mathbf{z}), \mathbf{z})$.

$\mathcal{D}_{[N]} := \{\mathcal{D}_1, \ldots, \mathcal{D}_N\}$. In other words, we decompose an offline task, learning with $\mathcal{D}$, into multiple offline sub-tasks, learning with $\mathcal{D}_n \in \mathcal{D}_{[N]}$ respectively. Then, for sub-task $n \in [1, N]$, we can perform behavior cloning (BC) to fit a parametric behavior policy $\beta_n : \mathcal{S} \times \mathcal{A} \to [0, 1]$ to model the corresponding offline subset $\mathcal{D}_n$:

$$\beta_n \leftarrow \arg\max_{\beta_n} \mathbb{E}_{(\mathbf{s}, \mathbf{a}) \sim \mathcal{D}_n} \big[ \log \beta_n(\mathbf{a}|\mathbf{s}) \big]. \tag{4}$$

Additionally, such a decomposition also comes with the benefit that it provides an avenue to exploit the hybrid modes in offline data $\mathcal{D}$, because that $\mathcal{D}$ is often collected using hybrid data-generating behavior policies [18], which suggests that fitting a single behavior policy may not be optimal to model the multiple modes of the offline data distribution (see Appendix A for empirical evidence). In real-world tasks, it is also often the case that offline data exhibit hybrid behaviors. Thus, to encourage the emergence of diverse sub-tasks that capture distinct behavior modes, we perform a simple task decomposition according to the returns of trajectories in $\mathcal{D}$, heuristically ensuring that trajectories in the same sub-task share similar returns and trajectories from different sub-tasks have distinct returns[4].

### 3.2 Task Embedding and Conservative MBO

**Naive MBO over behavior policies.** Benefiting from the above offline task decomposition, we can conduct MBO over a set of input-score $(\mathbf{x}, y)$ pairs, where we model the (parameters of) behavior policies $\beta_n$ as the design input $\mathbf{x}$ and the corresponding expected returns at initial states $J(\beta_n)$ as the score $y$. Note that, ideally, evaluating behavior policy $\beta_n$, *i.e.*, calculating $J(\beta_n) \equiv \mathbb{E}_{\mathbf{s}_0}\big[V^{\beta_n}(\mathbf{s}_0)\big]$, with subset $\mathcal{D}_n$ will never trigger the overestimation of values in the inner-level optimization. By introducing a score model $f : \Pi \to \mathbb{R}$ and setting it as the transfer information "[?]" in Q1, we can then perform outer-level policy inference with

$$\pi^*(\mathbf{a}|\mathbf{s}) \leftarrow \arg\max_{\pi} f(\pi), \text{ where } f = \arg\min_{f} \mathbb{E}_n \Big[ \big( f(\beta_n) - J(\beta_n) \big)^2 \Big]. \tag{5}$$

However, directly performing optimization-based inference (outer-level optimization), $\max_{\pi} f(\pi)$, will quickly find an invalid input for which the learned score model $f$ outputs erroneously large values (Q2). Furthermore, it is particularly severe if we perform the inference directly over the parameters of policy networks, accounting for the fact that the parameters of behavior policies lie on a narrow manifold in a high-dimensional parametric space.

**Task embedding.** To enable feasible out-level policy inference, we propose to decouple MBO techniques from high-dimensional policy parameters. We achieve this by learning a latent embedding space $\mathcal{Z}$ with an information bottleneck ($\dim(\mathcal{Z}) \ll \min(N, \dim(\Pi))$). As long as the high-dimensional parameters of behavior policies can be inferred from the embedding $\mathbf{z} \in \mathcal{Z}$, we can use $\mathbf{z}$ to represent the corresponding sub-task (or the behavior policy). Formally, we learn a deterministic task embedding[5] $\phi : \mathbb{R}^N \to \mathcal{Z}$ and a contextual behavior policy $\beta : \mathcal{S} \times \mathcal{Z} \times \mathcal{A} \to [0, 1]$ that replaces

---

[4]See more data decomposition rules in Appendix A.2.

[5]We feed the one-hot encoding of the sub-task specification ($n = 1, \ldots, N$) into the embedding network $\phi$. We write the embedding function as $\phi(\mathbf{z}|n)$ instead of $\phi(n)$ to emphasize that the output of $\phi$ is $\mathbf{z}$.

$N$ separate behavior policies in Equation 4:

$$\beta(\mathbf{a}|\mathbf{s},\mathbf{z}), \phi(\mathbf{z}|n) \leftarrow \arg\max_{\beta,\phi} \ \mathbb{E}_{\mathcal{D}_n \sim \mathcal{D}_{[N]}} \mathbb{E}_{(\mathbf{s},\mathbf{a})\sim\mathcal{D}_n} \left[ \log\beta(\mathbf{a}|\mathbf{s},\phi(\mathbf{z}|n)) \right], \tag{6}$$

**Conservative MBO.** In principle, by substituting the learned task embedding $\phi(\mathbf{z}|n)$ and contextual behavior policy $\beta(\mathbf{a}|\mathbf{s},\mathbf{z})$ into Equation 5, we can then conduct MBO over the embedding space: first learning $f : \mathcal{Z} \rightarrow \mathbb{R}$ with $\min_f \mathbb{E}_{n,\phi(\mathbf{z}|n)} \left[ (f(\mathbf{z}) - J(\beta_n))^2 \right]$, and then setting the optimal embedding with $\mathbf{z}^* = \arg\max_{\mathbf{z}} f(\mathbf{z})$ and the corresponding optimal policy with $\pi^*(\mathbf{a}|\mathbf{s}) = \beta(\mathbf{a}|\mathbf{s},\mathbf{z}^*)$. However, we must deliberate a new OOD issue in the $\mathcal{Z}$-space, stemming from the original distribution shift in the parametric space when directly optimizing Equation 5.

Motivated by the energy model [38] and the conservative regularization [35], we introduce a conservative score model objective, additionally regularizing the scores of OOD embeddings $\mu(\mathbf{z})$:

$$f \leftarrow \arg\min_f \ \mathbb{E}_{n,\phi(\mathbf{z}|n)} \left[ \left( f(\mathbf{z}) - J(\beta_n) \right)^2 \right], \ \text{s.t.} \ \mathbb{E}_{\mu(\mathbf{z})}\left[f(\mathbf{z})\right] - \mathbb{E}_{n,\phi(\mathbf{z}|n)}\left[f(\mathbf{z})\right] \leq \eta, \tag{7}$$

where we set $\mu(\mathbf{z})$ to be the uniform distribution over $\mathcal{Z}$-space. Intuitively, as long as the scores of OOD embeddings $\mathbb{E}_{\mu(\mathbf{z})}\left[f(\mathbf{z})\right]$ is lower than that of in-distribution embeddings $\mathbb{E}_{n,\phi(\mathbf{z}|n)}\left[f(\mathbf{z})\right]$ (up to a threshold $\eta$), conducting embedding inference with $\mathbf{z}^* = \arg\max_{\mathbf{z}} f(\mathbf{z})$ would produce the best and confident solution, avoiding towards OOD embeddings that are far away from the training set.

Now we have reframed the non-iterative bi-level offline RL problem as one of offline MBO: in the inner-level optimization (Q1), we set the practical choice for "?" as the learned score model $f$ (A1); in the outer-level optimization (Q2), we introduce task embedding and conservative regularization to avoid over-optimistic exploitation when exploiting $f$ for policy/embedding inference (A2). In the next section, we will show how to slightly change the form of the score model $f$, so as to leverage the sequential characteristic (loop in Figure 1) of RL tasks and answer the left Q3.

### 3.3 Test-time Adaptation

Recalling that we update $f(\mathbf{z})$ to regress the value at initial states $\mathbb{E}_{\mathbf{s}_0}\left[V^{\beta_n}(\mathbf{s}_0)\right]$ in Equation 7, we then conduct outer-level inference with $\mathbf{z}^* = \arg\max_{\mathbf{z}} f(\mathbf{z})$ and rollout the $z^*$-conditioned policy $\pi^*(\mathbf{a}|\mathbf{s}) := \beta(\mathbf{a}|\mathbf{s},\mathbf{z}^*)$ until the end of testing/deployment rollout episode. In essence, such an inference can produce a confident extrapolation over the distribution of behavior policy embeddings. Going beyond the outer-level policy/embedding inference only at the initial states, we propose that we can benefit from performing inference at any rollout state in deployment (A3).

To enable test-time adaptation, we model the score model with $f : \mathcal{S} \times \mathcal{A} \times \mathcal{Z} \rightarrow \mathbb{R}$, taking a state-action as an extra input. Then, we encourage the score model to regress the values of behavior policies over all state-action pairs, $\min_f \mathbb{E}_{n,\phi(\mathbf{z}|n)}\mathbb{E}_{(\mathbf{s},\mathbf{a})\sim\mathcal{D}_n} \left[ \left( f(\mathbf{s},\mathbf{a},\mathbf{z}) - Q^{\beta_n}(\mathbf{s},\mathbf{a}) \right)^2 \right]$.
For simplicity, instead of learning a separate value function $Q^{\beta_n}$ for each behavior policy, we learn the score model directly with the TD-error used for learning the value function $Q^{\beta_n}(\mathbf{s},\mathbf{a})$ as in Equation 1, together with the conservative regularization in Equation 7:

$$f \leftarrow \arg\min_f \mathbb{E}_{\mathcal{D}_n \sim \mathcal{D}_{[N]}} \mathbb{E}_{(\mathbf{s},\mathbf{a},\mathbf{s}',\mathbf{a}')\sim\mathcal{D}_n} \left[ \left( R(\mathbf{s},\mathbf{a}) + \gamma\bar{f}(\mathbf{s}',\mathbf{a}',\phi(\mathbf{z}|n)) - f(\mathbf{s},\mathbf{a},\phi(\mathbf{z}|n)) \right)^2 \right], \tag{8}$$

$$\text{s.t.} \ \mathbb{E}_{n,\mu(\mathbf{z})}\mathbb{E}_{\mathbf{s}\sim\mathcal{D}_n,\mathbf{a}\sim\beta(\mathbf{a}|\mathbf{s},\mathbf{z})}\left[f(\mathbf{s},\mathbf{a},\mathbf{z})\right] - \mathbb{E}_{n,\phi(\mathbf{z}|n)}\mathbb{E}_{\mathbf{s}\sim\mathcal{D}_n,\mathbf{a}\sim\beta(\mathbf{a}|\mathbf{s},\mathbf{z})}\left[f(\mathbf{s},\mathbf{a},\mathbf{z})\right] \leq \eta,$$

where $\bar{f}$ denotes a target network and we update the target $\bar{f}$ with soft updates: $\bar{f} = (1 - \upsilon)\bar{f} + \upsilon f$.

In test-time, we thus can dynamically adapt the outer-level optimization, setting the inferred optimal policy $\pi^*(\mathbf{a}|\mathbf{s}) = \beta(\mathbf{a}|\mathbf{s},\mathbf{z}^*(\mathbf{s}))$, where $z^*(\mathbf{s}) = \arg\max_z f\left(\mathbf{s},\beta(\mathbf{a}|\mathbf{s},\mathbf{z}),\mathbf{z}\right)$. Specifically, at any state $\mathbf{s}$ in the test-time, we can perform gradient ascent to find the optimal behavior embedding: $z^*(\mathbf{s}) = \mathbf{z}_K^\circ(\mathbf{s}) := \text{GradAscent}(\mathbf{s},\mathbf{z}_0^\circ,K)$, where $\mathbf{z}_0^\circ$ is the starting point and for $k = 0,1\ldots,K-1$,

$$\mathbf{z}_{k+1}^\circ(\mathbf{s}) \leftarrow \mathbf{z}_k^\circ(\mathbf{s}) + \eta\nabla_{\mathbf{z}}f(\mathbf{s},\beta(\mathbf{a}|\mathbf{s},\mathbf{z}),\mathbf{z}))|_{\mathbf{z}=\mathbf{z}_k^\circ}. \tag{9}$$

### 3.4 Connection to Prior Non-iterative Offline Methods

In Table 1, we summarize the comparison with prior representative non-iterative offline RL methods. Intuitively, our DROP (using returns to decompose $\mathcal{D}$) is similar in spirit to F-BC[6] and RvS-R [14],

---

[6]F-BC performs BC over filtered trajectories with high returns.

Table 1: **Comparison of non-iterative bi-level offline RL methods**, where we use $\mathsf{R}(\tau)$ to represent the return of a sampling trajectory $\tau$ and use $\mathsf{R}(\mathbf{s}, \mathbf{a})$ to represent the expected return of trajectories starting from $(\mathbf{s}, \mathbf{a})$. The checkmark in `A2` indicates whether the exploitation to "⍰" is regularized for outer-level optimization and that in `A3` indicates whether test-time adaptation is supported.

| | Inner-level | Outer-level | "⍰" in `A1` | `A2` | `A3` |
|---|---|---|---|---|---|
| F-BC | BC over filtered $\tau$ with high $\mathsf{R}(\tau)$ | — | $\pi(\mathbf{a}\|\mathbf{s})$ | ✗ | ✗ |
| RvS-R [14] | $\min_\pi -\mathbb{E}\left[\log \pi(\mathbf{a}\|\mathbf{s}, \mathsf{R}(\tau))\right]$ | handcraft $\mathsf{R}_{\text{target}}$ | $\pi(\mathbf{a}\|\mathbf{s}, \cdot)$ | ✗ | ✗ |
| Onestep [6] | $\min_Q \mathcal{L}(Q(\mathbf{s}, \mathbf{a}), \mathsf{R}(\mathbf{s}, \mathbf{a}))$ | $\arg\max_{\mathbf{a}} Q_\beta(\mathbf{s}, \beta(\mathbf{a}\|\mathbf{s}))$ | $Q_\beta(\mathbf{s}, \mathbf{a})$ | ✗ | ✔ |
| COMs [62] | $\min_f \mathcal{L}(f(\beta_\tau), \mathsf{R}(\tau))$ | $\arg\max_\beta f(\beta)$ | $f(\beta)$ | ✔ | ✗ |
| DROP (ours) | $\min_f \mathcal{L}(f(\mathbf{s}, \mathbf{a}, \mathbf{z}), \mathsf{R}(\mathbf{s}, \mathbf{a}, \mathbf{z}))$ | $\arg\max_{\mathbf{z}} f(\mathbf{s}, \beta(\mathbf{a}\|\mathbf{z}), \mathbf{z})$ | $f(\mathbf{s}, \mathbf{a}, \mathbf{z})$ | ✔ | ✔ |

both of which use return $\mathsf{R}(\tau)$ to guide the inner-level optimization. However, both F-BC and RvS-R leave `Q2` unanswered. In outer-level, F-BC can not enable policy extrapolation, which heavily relies on the data quality in offline tasks. RvS-R needs to handcraft a target return (as the contextual variable for $\pi(\mathbf{a}\|\mathbf{s}, \cdot)$), which also probably triggers the potential distribution mismatch[7] between the hand-crafted contextual variable (*i.e.*, the desired return) and the actual rollout return induced by the learned policy when conditioning on the given contextual variable.

Diving deeper into the bi-level optimization, we can also find DROP combines the advantages of Onestep [6] and COMs [62], where Onestep answers `Q3` and performs outer-level optimization in action space ($\arg\max_{\mathbf{a}}$), COMs answers `Q2` and performs that in parameter space ($\arg\max_\beta$), while our DROP answers both `Q2` and `Q3` and performs that in embedding space ($\arg\max_{\mathbf{z}}$). Specifically, DROP thus allows us to (`A2`) conduct safe extrapolation in outer-level and (`A3`) perform test-time adaptation in testing rather than simply conducting outer-level optimization at initial states as in COMs.

### 3.5 Practical Implementation

| **Algorithm 1** DROP (Training) | **Algorithm 2** DROP (Testing/Deployment) |
|---|---|
| **Require:** Dataset of trajectories: $\mathcal{D} = \{\tau\}$. | **Require:** Env, $\beta(\mathbf{a}\|\mathbf{s}, \mathbf{z})$, and $f(\mathbf{s}, \mathbf{a}, \mathbf{z})$. |
| 1: Decompose $\mathcal{D}$ into $N$ sub-sets $\mathcal{D}_{[N]}$. | 1: $\mathbf{s}_0 = $ Env.Reset(). |
| 2: Initialize $\phi(\mathbf{z}\|n)$, $\beta(\mathbf{a}\|\mathbf{s}, \mathbf{z})$, and $f(\mathbf{s}, \mathbf{a}, \mathbf{z})$. | 2: **while** not done **do** |
| 3: **while** not converged **do** | 3:     *Test-time adaptation:* |
| 4:     Sample a sub-task: $\mathcal{D}_n \sim \mathcal{D}_{[N]}$. |     $\mathbf{z}^*(\mathbf{s}_t) = \arg\max_{\mathbf{z}} f(\mathbf{s}_t, \beta(\mathbf{a}_t\|\mathbf{s}_t, \mathbf{z}), \mathbf{z})$. |
| 5:     Learn $\phi$, $\beta$, and $f$ with Equations 6 and 8. | 4:     Sample action: $\mathbf{a}_t \sim \beta(\mathbf{a}_t\|\mathbf{s}_t, \mathbf{z}^*(\mathbf{s}_t))$. |
| 6: **end while** | 5:     Step Env: $\mathbf{s}_{t+1} \sim P(\mathbf{s}_{t+1}\|\mathbf{s}_t, \mathbf{a}_t)$. |
| **Return:** $\beta(\mathbf{a}\|\mathbf{s}, \mathbf{z})$ and $f(\mathbf{s}, \mathbf{a}, \mathbf{z})$. | 6: **end while** |

We now summarize the DROP algorithm (see Algorithm 1 for the training phase and Algorithm 2 for the testing phase). During training (inner-level optimization), we alternate between updating $\phi(\mathbf{z}\|n)$, $\beta(\mathbf{a}\|\mathbf{s}, \mathbf{z})$, and $f(\mathbf{s}, \mathbf{a}, \mathbf{z})$, wherein we update $\phi$ with both maximum likelihood loss and TD-error loss in Equations 6 and 8. During testing (outer-level optimization), we use the gradient ascent as specified in Equation 9 to infer the optimal embedding $z^*$. Instead of simply sampling a single starting point $\mathbf{z}_0^\circ$, we choose $N$ starting points corresponding to all the embeddings $\{\mathbf{z}_n\}$ of sub-tasks, and then choose the optimal $\mathbf{z}^*(\mathbf{s})$ from those *updated embeddings* for which the learned $f$ outputs the highest score, *i.e.*, $\mathbf{z}^*(\mathbf{s}) = \arg\max_{\mathbf{z}} f(\mathbf{s}, \beta(\mathbf{a}\|\mathbf{s}, \mathbf{z}), \mathbf{z})$ s.t. $\mathbf{z} \in \{\text{GradAscent}(\mathbf{s}, \mathbf{z}_n, K)|n = 1, \ldots, N\}$. Then, we sample action from $\pi^*(\mathbf{a}\|\mathbf{s}) := \beta(\mathbf{a}\|\mathbf{s}, \mathbf{z}^*(\mathbf{s}))$.

## 4 Related Work

**Offline RL.** In offline RL, learning with offline data is prone to exploiting OOD state-actions and producing over-estimated values, which makes vanilla iterative policy/value optimization challenging. To eliminate the problem, a number of methods have been explored, in essence, by either *introducing a policy/value regularization in the iterative loop* or *trying to eliminate the iterative loop itself*.

*Iterative methods:* Sticking with the normal iterative updates in RL, offline policy regularization methods aim to keep the learning policy to be close to the behavior policy under a probabilistic

---

[7]For more illustrative examples w.r.t. this mismatch, we refer the reader to Figure 6 of the RvS-R paper [14].

distance [8, 19, 30, 33, 45, 46, 52, 54, 58, 67, 73]. Some works also conduct implicit policy regularization with variants of importance sampling [39, 47, 51]. Besides regularizing policy, it is also feasible to constrain the substitute value function in the iterative loop. Methods constraining the value function aim at mitigating the over-estimation, which typically introduces pessimism to the prediction of the Q-values [9, 27, 35, 41, 48, 49] or penalizes the value with an uncertainty quantification [3, 5, 56, 68], making the value for OOD state-actions more conservative. Similarly, another branch of model-based methods [29, 71, 72, 57] also perform iterative bi-level updates, alternating between regularized evaluation and improvement. Different from this iterative paradigm, DROP only evaluates values of behavior policies in the inner-level optimization, avoiding the potential overestimation for values of learning policies and eliminating error propagation between two levels.

*Non-iterative methods:* Another complementary line of work studies how to eliminate the iterative updates, which simply casts RL as a *weighted* or *conditional* imitation learning problem (Q1). Derived from the behavior-regularization RL [22, 64], *the former* conducts weighted behavior cloning: first learning a value function for the behavior policy, then weighing the state-action pairs with the learned values or advantages [1, 12, 66, 54]. Besides, some works propose an implicit behavior policy regularization that similarly does not estimate values of new learning policies: initializing the learning policy with a behavior policy [50] or performing only a "one-step" update (policy improvement) over the behavior policy [20, 24, 75]. For *the latter*, this branch method typically builds upon the hindsight information matching [4, 15, 28, 36, 44, 43, 55, 65], assuming that the future trajectory information can be useful to infer the middle decision that leads to the future and thus relabeling the trajectory with the reached states or returns. Due to the simplicity and stability, RvS-based methods advocate for learning a goal-conditioned or reward-conditioned policy with supervised regression [10, 13, 14, 21, 25, 42, 60, 69]. However, these works do not fully exploit the non-iterative framework and fail to answer the raised questions, which either does not regularize the inner-level optimization when exploiting "⍰" in outer-level (Q2), or does not support test-time adaptation (Q3).

**Offline MBO.** Similar to offline RL, the main challenge of MBO is to reason about uncertainty and OOD values [7, 16], since a direct gradient-ascent against the learned score model can easily produce invalid inputs that are falsely and highly scored. To counteract the effect of model exploitation, prior works introduce various techniques, including normalized maximum likelihood estimation [17], model inversion networks [32], local smoothness prior [70], and conservative objective models (COMs) [62]. Compared to COMs, DROP shares similarities with the conservative model, but instantiates on the embedding space instead of the parameter space. Such difference is nontrivial, not only because DROP avoids the adversarial optimization employed in COMs, but also because DROP allows test-time adaptation, enabling dynamical inference across states at testing.

## 5 Experiments

In this section, we present our empirical results. We first give examples to illustrate the test-time adaptation. Then we evaluate DROP against prior offline RL algorithms on the D4RL benchmark. Finally, we provide the computation cost regarding the test-time adaptation protocol. For more offline-to-online fine-tuning results, ablation studies w.r.t. the decomposition rules and the conservative regularization, and training details on the hyper-parameters, we refer the readers to the appendix.

**Illustration of test-time adaptation.** To better understand the test-time adaptation of DROP, we include four comparisons that exhibit different embedding inference rules at testing/deployment:

(1) DROP-Best: At initial state $\mathbf{s}_0$, we choose the best embedding from those embeddings of behavior policies, $\mathbf{z}_0^*(\mathbf{s}_0) = \arg\max_z f(\mathbf{s}_0, \beta(\mathbf{a}_0|\mathbf{s}_0, \mathbf{z}), \mathbf{z})$ s.t. $\mathbf{z} \in \mathbf{z}_{[N]} := \{\mathbf{z}_1, \ldots, \mathbf{z}_N\}$, and keep this embedding fixed for the entire episode, *i.e.*, setting $\pi^*(\mathbf{a}_t|\mathbf{s}_t) = \beta(\mathbf{a}_t|\mathbf{s}_t, \mathbf{z}_0^*(\mathbf{s}_0))$.

(2) DROP-Grad: At initial state $\mathbf{s}_0$, we conduct inference (gradient ascent on starting point $\mathbf{z}_0^*(\mathbf{s}_0)$) with $\mathbf{z}^*(\mathbf{s}_0) = \arg\max_z f(\mathbf{s}_0, \beta(\mathbf{a}_0|\mathbf{s}_0, \mathbf{z}), \mathbf{z})$, and keep $\mathbf{z}^*(\mathbf{s}_0)$ fixed throughout the test rollout.

(3) DROP-Best-Ada: We adapt the policy by setting $\pi^*(\mathbf{a}_t|\mathbf{s}_t) = \beta(\mathbf{a}_t|\mathbf{s}_t, \mathbf{z}_0^*(\mathbf{s}_t))$, where we choose the best embedding $\mathbf{z}_0^*(\mathbf{s}_t)$ directly from those embeddings of behavior policies for which the score model outputs the highest score, *i.e.*, $\mathbf{z}_0^*(\mathbf{s}_t) = \arg\max_{\mathbf{z}} f(\mathbf{s}_t, \beta(\mathbf{a}_t|\mathbf{s}_t, \mathbf{z}), \mathbf{z})$ s.t. $\mathbf{z} \in \mathbf{z}_{[N]}$.

(4) DROP-Grad-Ada (as described in Section 3.5): We set $\pi^*(\mathbf{a}_t|\mathbf{s}_t) = \beta(\mathbf{a}_t|\mathbf{s}_t, \mathbf{z}^*(\mathbf{s}_t))$ and choose the best embedding $\mathbf{z}^*(\mathbf{s}_t)$ from those *updated embeddings* of behavior policies, *i.e.*, $\mathbf{z}^*(\mathbf{s}_t) = \arg\max_{\mathbf{z}} f(\mathbf{s}_t, \beta(\mathbf{a}_t|\mathbf{s}_t, \mathbf{z}), \mathbf{z})$ s.t. $\mathbf{z} \in \{\mathrm{GradAscent}(\mathbf{s}_t, \mathbf{z}_n, K)|n = 1, \ldots, N\}$.

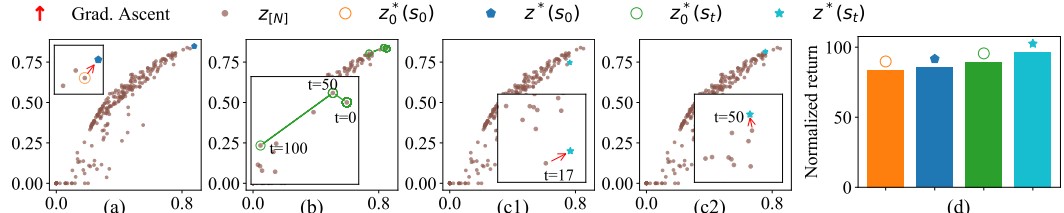

Figure 3: **Visualization of the embedding inference (a, b, c1, c2) and performance comparison (d).** $\mathbf{z}_{[N]}$ denotes embeddings of all behavior policies. $\mathbf{z}_0^*(\mathbf{s}_0)$, $\mathbf{z}^*(\mathbf{s}_0)$, $\mathbf{z}_0^*(\mathbf{s}_t)$, and $\mathbf{z}^*(\mathbf{s}_t)$ denote the selected embeddings in DROP- Best, Grad, Best-Ada, and Grad-Ada implementations respectively.

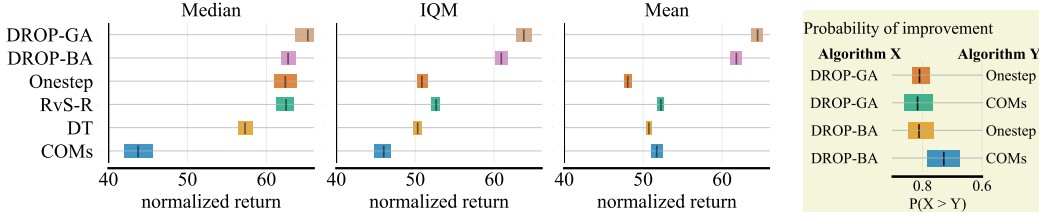

Figure 4: (*Left*) **Aggregate metrics [2] on 12 D4RL tasks**. Higher median, IQM, and mean scores are better. (*Right*) Each row shows the probability of improvement that the algorithm X on the left outperforms algorithm Y on the right. The CIs are estimated using the percentile bootstrap (95%) with stratified sampling. For all results of our method, we average the normalized returns across 5 seeds; for each seed, we run 10 evaluation episodes. (GA: Grad-Ada. BA: Best-Ada.)

In Figure 3, we visualize the four different inference rules and report the corresponding performance in the halfcheetah-medium-expert task [18]. In Figure 3 (a), we set the starting point as the best embedding $\mathbf{z}_0^*(\mathbf{s}_0)$ in $\mathbf{z}_{[N]}$, and perform gradient ascent to find the optimal $\mathbf{z}_0^*(\mathbf{s}_0)$ for DROP-Grad. In Figure 3 (b), we can find that at different time steps, DROP-Best-Ada chooses different embeddings for $\beta(\mathbf{a}_t|\mathbf{s}_t, \cdot)$. Intuitively, performing such a dynamical embedding inference enables us to combine different embeddings, *thus stitching behavior policies at different states*. Further, in Figure 3 (c1, c2), we find that performing additional inference (with gradient ascent) in DROP-Grad-Ada allows to extrapolate beyond the embeddings of behavior policies, and *thus results in sequential composition of new embeddings (policies) across different states*. For practical impacts of these different inference rules, we provide the performance comparison in Figure 3 (d), where we can find that performing gradient-based optimization (*-Grad-*) outperforms the natural selection among those embeddings of behavior policies/sub-tasks (*-Best-*), and rollout with adaptive embedding inference (DROP-*-Ada) outperforms that with fixed embeddings (DROP-Best and DROP-Grad). In subsequent experiments, if not explicitly stated, DROP takes the DROP-Grad-Ada implementation by default.

**Empirical performance on benchmark tasks.** We evaluate DROP on a number of tasks from the D4RL dataset and make comparisons with prior non-iterative offline RL counterparts[8].

As suggested by Agarwal et al. [2], we provide the aggregate comparison results to account for the statistical uncertainty. In Figure 4, we report the median, IQM [2], and mean scores of DROP and prior non-iterative offline baselines in 12 D4RL tasks (see individual comparison results in Appendix C). We can find our DROP (Best-Ada and Grad-Ada) consistently outperforms prior non-iterative baselines in both median, IQM, and mean metrics. Compared to the most related baseline Onestep and COMs (see Table 1), we provide the probability of performance improvement in Figure 4 *right*. We can find that DROP-Grad-Ada shows a robust performance improvement over Onestep and COMs, *with an average improvement probability of more than 80%*.

**Comparison with latent policy methods.** Note that one additional merit of DROP is that it naturally accounts for hybrid modes in $\mathcal{D}$ by conducting task decomposition in inner-level, we thus compare DROP to latent policy methods (PLAS [74] and LAPO [11]) that use conditional variational autoencoder (CVAE) to model offline data and also account for multi-modes in offline data. Essentially, both our DROP and baselines (PLAS and LAPO) learn a latent policy in the inner-level optimization, except that we adopt the non-iterative bi-level learning while baselines are instantiated

---

[8]Due to the page limit, we provide the comparison with prior iterative baselines in Appendix C.

Table 2: **Comparison in AntMaze and Gym-MuJoCo domains (v2).** Results are averaged over 5 seeds; for each seed, we run 10 evaluation episodes. u: umaze (antmaze). um: umaze-medium. ul: umaze-large. p: play. d: diverse. wa: walker2d. ho: hopper. ha: halfcheetah. r: random. m: medium.

|          | u     | u-d   | um-p  | um-d  | ul-p  | ul-d  | wa-r  | ho-r  | ha-r  | wa-m  | ho-m  | ha-m  | mean  |
|----------|-------|-------|-------|-------|-------|-------|-------|-------|-------|-------|-------|-------|-------|
| CQL      | 74.0  | 84.0  | 61.2  | 53.7  | 15.8  | 14.9  | -0.2  | 8.3   | 22.2  | **82.1** | 71.6  | 49.8  | 44.8  |
| IQL      | 87.5  | 62.2  | 71.2  | 70.0  | 39.6  | 47.5  | **5.4** | 7.9   | 13.1  | 77.9  | 65.8  | 47.8  | 49.7  |
| PLAS     | 70.7  | 45.3  | 16.0  | 0.7   | 0.7   | 0.3   | 9.2   | 6.7   | 26.5  | 75.5  | 51.0  | 44.5  | 28.9  |
| LAPO     | 86.5  | 80.6  | 68.5  | 79.2  | 48.8  | **64.8** | 1.3   | **23.5** | 30.6  | 80.8  | 51.6  | 46.0  | 55.2  |
| DROP     | **90.5** | **92.2** | **74.1** | **82.9** | **57.2** | 63.3  | 5.2   | 20.8  | **32.0** | 82.1  | **74.9** | **52.4** | **60.6** |
| DROP std | ±2.4  | ±1.7  | ±3.9  | ±3.5  | ±5.5  | ±2.4  | ±1.6  | ±0.3  | ±2.5  | ±5.2  | ±2.8  | ±2.2  |       |

under the iterative paradigm. By answering Q3, DROP permits test-time adaptation, enabling us to dynamically stitch "skills"(latent behaviors/embeddings as shown in Figure 3) and thus encouraging high-level abstract exploitation in testing. However, the aim of introducing the latent policy in PLAS and LAPO is to regularize the inner-level optimization, which fairly answers Q2 in the iterative offline counterpart but can not provide the potential benefit (test-time adaptation) by answering Q3.

Note in our naive DROP implementation, we heuristically use the return to conduct task decomposition (motivated by RvS-R), while PLAS and LAPO take each trajectory as a sub-task and use CVAE to learn the latent policy. For a fair comparison, here we also adopt CVAE to model the offline data and afterward take the learned latent embedding in CVAE as the embedding of behaviors, instead of conducting return-guided task decomposition. We provide implementation details (DROP+CVAE) in Appendix A.4 and the comparison results in Table 2 (taking DROP-Grad-Ada implementation). We can observe that our DROP consistently achieves better performance than PLAS in all tasks, and performs better or comparably to LAPO in 10 out of 12 tasks. Additionally, we also provide the results of CQL and IQL [31]. We can find that DROP also leads to significant improvement in performance, consistently demonstrating the competitive performance of DROP against state-of-the-art offline iterative/non-iterative baselines.

**Computation cost.** One limitation of DROP is that conducting test-time adaptation at each state is usually expensive for inference time. To reduce the computation cost, we find that we can resample (infer) the best embedding $\mathbf{z}^*$ after a certain time interval, *i.e.*, $\mathbf{z}^*$ does not necessarily need to be updated at each testing state. In Figure 5, we plot the average inference time versus different models and DROP implementations with varying inference time intervals. We see that compared to Diffuser (one offline RL method that also supports test-time adaptation, see quantitative performance comparison in Table 4) [26], DROP can achieve significantly lower computation costs. In Figure 5 *top-right*, we also illustrate the trade-off between performance and runtime budget as we vary the inference time intervals. We find that by choosing a

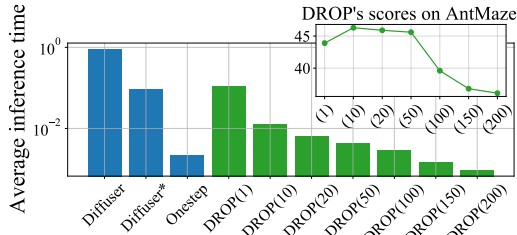

Figure 5: **Computation cost across different models and DROP implementations with varying inference time intervals.** The number in parenthesis denotes the inference time intervals. In *top-right*, we also provide the average performance of DROP on AntMaze tasks when using different inference time intervals. (Diffuser* denotes the warm-start implementation of Diffuser.)

suitable time interval, we can reduce the inference cost with only a modest drop in performance: DROP(50) brings less computational burden while providing stable performance.

**Comparison to a "distilled" DROP implementation.** Our answer to Q3 is that we conduct outer-level optimization in the testing phase. Similarly, we can design a "distilled" DROP implementation: we keep the same treatment as DROP for Q1 and Q2, but we perform outer-level optimization on the existing offline data instead of on the testing states, *i.e.*, conducting optimization in $\mathbf{z}$-space for all states in the dataset. Then, we can "distill" the resulting contextual policy and the inferred

Table 3: **Comparison to "distilled" DROP (D-DROP),** using Grad-Ada and CVAE implementation.

|        | wa-r  | ho-r  | ha-r  | wa-m  | ho-m  | ha-m  |
|--------|-------|-------|-------|-------|-------|-------|
| CQL    | -0.2  | 8.3   | 22.2  | 82.1  | 71.6  | 49.8  |
| IQL    | **5.4** | 7.9   | 13.1  | 77.9  | 65.8  | 47.8  |
| D-DROP | 4.5   | 18.9  | 27.8  | **84.2** | 67.1  | 50.9  |
| DROP   | 5.2   | **20.8** | **32.0** | 82.1  | **74.9** | **52.4** |

$\mathbf{z}^*$ into a fixed rollout policy. As shown in Table 3, we can see that such a "distilled" DROP implementation achieves competitive performance, approaching DROP's overall score and slightly outperforming CQL and IQL, which further supports the superiority afforded by the non-iterative offline RL paradigm versus the iterative one.

**Comparison to adaptive baselines.** One merit of DROP resides in its test-time adaptation ability across rollout states. Thus, here we compare DROP with two offline RL methods that also support test-time adaptation: 1) APE-V [23] learns an uncertainty-adaptive policy and dynamically updates a con-

Table 4: **Comparison to adaptive baselines (APE-V and Diffuser).** DROP uses Grad-Ada and CVAE implementation.

|          | wa-mr | ho-mr | ha-mr | wa-me | ho-me | ha-me |
|----------|-------|-------|-------|-------|-------|-------|
| APE-V    | 82.9  | **98.5** | **64.6** | **110.0** | 105.7 | 101.4 |
| Diffuser | 70.6  | 93.6  | 37.7  | 106.9 | 103.3 | 88.9  |
| DROP     | **83.5** | 96.3 | 50.9 | 109.3 | **107.2** | **102.2** |

textual belief vector at test states, and 2) Diffuser [26] employs a score/return function to guide the diffusion denoising process, *i.e.*, score-guided action sampling. As shown in Table 4, we can see that in medium-replay (mr) and medium-expert (me) domains (v2), DROP can achieve comparable results to APE-V, with a clear improvement over Diffuser.

## 6 Discussion

In this work, we introduce non-iterative bi-level offline RL, and based on this paradigm, we raise three questions (Q1, Q2, and Q3). To answer that, we reframe the offline RL problem as one of MBO and learn a score model (A1), introduce embedding learning and conservative regularization (A2), and propose test-time adaptation in testing (A3). We evaluate DROP on various tasks, showing that DROP gains comparable or better performance compared to prior methods.

**Limitations.** DROP also has several limitations. First, the offline data decomposition dominates the following bi-level optimization, and thus choosing a suitable decomposition rule is a crucial requirement for policy inference (see experimental analysis in Appendix A.2). An exciting direction for future work is to study generalized task decomposition rules (*e.g.*, our DROP+CVAE implementation). Second, we find that when the number of sub-tasks is too large, the inference is unstable, where adjacent checkpoint models exhibit larger variance in performance (such instability also exists in prior offline RL methods, discovered by Fujimoto and Gu [19]). One natural approach to this instability is conducting online fine-tuning (see Appendix A.3 for our empirical studies).

Going forward, we believe our work suggests a feasible alternative for generalizable offline robotic learning: by decomposing a single robotic dataset into multiple subsets, offline policy inference can benefit from performing MBO and the test-time adaptation protocol.

## Acknowledgments and Disclosure of Funding

We sincerely thank the anonymous reviewers for their insightful suggestions. This work was supported by the National Science and Technology Innovation 2030 - Major Project (Grant No. 2022ZD0208800), and NSFC General Program (Grant No. 62176215).

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

# Appendix

## A More Experiments

### A.1 Single Behavior Policy v.s. Multiple Behavior Policies

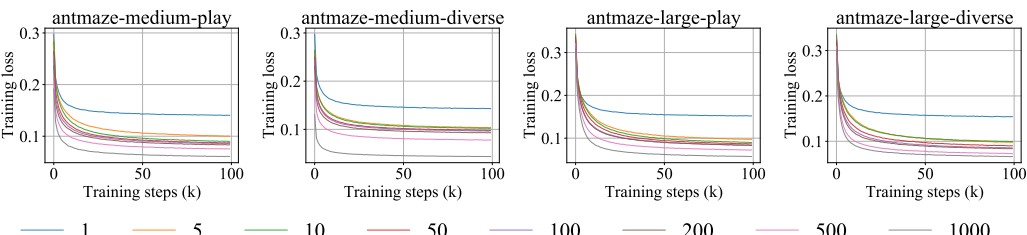

Figure 6: Learning curves of behavior cloning on AntMaze suites (*-v2) in D4RL, where the x-axis denotes the training steps, and the y-axis denotes the training loss. The number $N$ in the legend denotes the number of sub-tasks. If $N = 1$, we learn a single behavior policy for the whole offline dataset.

In Figure 6, we provide empirical evidence that learning a single behavior policy (using BC) is not sufficient to characterize the whole offline dataset, and multiple behavior policies (conducting task decomposition) deliver better resilience to characterize the offline data than a single behavior policy.

### A.2 Decomposition Rules

In DROP algorithm, we explicitly decompose an offline task into multiple sub-tasks, over which we then reframe the offline policy learning problem as one of offline model-based optimization. In this section, we discuss three different designs for the task decomposition rule.

**Random**$(N, M)$**:** We decomposition offline dataset $\mathcal{D} := \{\tau\}$ into $N$ subsets, each of which contains at most $M$ trajectories that are randomly sampled from the offline dataset.

**Quantization**$(N, M)$**:** Leveraging the returns of trajectories in offline data, we first quantize offline trajectories into $N$ bins, and then randomly sample at most $M$ trajectories (as a sub-task) from each bin. Specifically, in the $i$-th bin, the quantized trajectories $\{\tau_i\}$ satisfy $R_{min} + \Delta * i < \text{Return}(\tau_i) \leq R_{min} + \Delta * (i + 1)$, where $\Delta = \frac{(R_{max} - R_{min})}{N}$, $\text{Return}(\tau_i)$ denotes the return of trajectory $\tau_i$, and $R_{max}$ and $R_{min}$ denote the maximum and minimum trajectory returns in the offline dataset respectively.

**Rank**$(N, M)$**:** We first rank the offline trajectories descendingly based on their returns, and then sequentially sample $M$ trajectories for each subset. (*We adopt this decomposition rule in main paper.*)

In Figure 7, we provide the comparison of the above three decomposition rules (see the selected number of sub-tasks and the number of trajectories in each sub-task in Table 9). We can find that across a variety of tasks, the decomposition rule has a fundamental impact on the subsequent model-based optimization. Across different tasks and different embedding inference rules, Random and Quantization decomposition rules tend to exhibit large performance fluctuations, which reveals the importance of choosing a suitable task decomposition rule. In our paper, we adopt the Rank decomposition rule, as it demonstrates a more robust performance shown in Figure 7. In Appendix A.4, we adopt the conditional variational auto-encoder (CVAE) to conduct automatic task decomposition (treating each trajectory in offline dataset as an individual task) and we find such implementation (DROP+CVAE) can further improve DROP's performance. In future work, we also encourage better decomposition rules to decompose offline tasks so as to enable more effective model-based optimization for offline RL tasks.

**Comparison with filtered behavior cloning.** We also note that the Rank decomposition rule leverages more high-quality (high-return) trajectories than the other two decomposition rules (Random and Quantization). Thus, a natural question to ask is, is the performance of Rank better than that of Random and Quantization due to the presence of more high-quality trajectories in the decomposed sub-tasks? That is, whether DROP (using the Rank decomposition rule) only conducts behavioral cloning over those high-quality trajectories, thus leading to better performance.

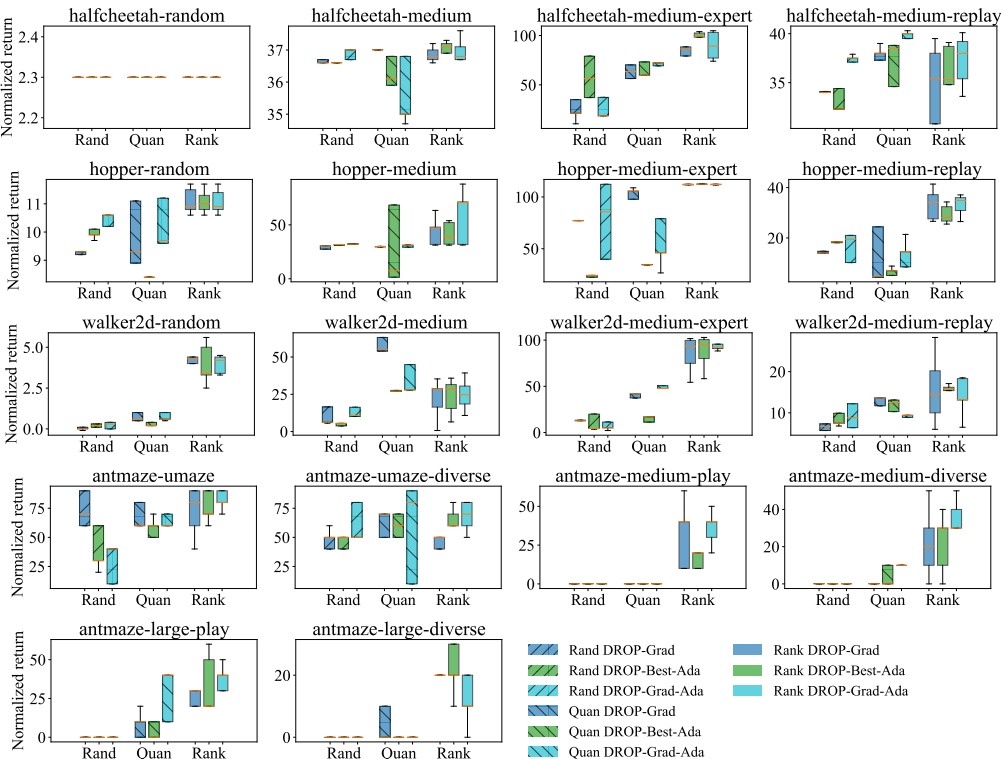

Figure 7: Comparison of three different decomposition rules on D4RL MuJoCo-Gym suite (*-v0) and AntMaze suite (*-v2), where "Rand", "Quan" and "Rank" denote the Random, Quantization, and Rank decomposition rules respectively. We can find across 18 tasks (AntMaze and MuJoCo-Gym suites) and 3 embedding inference methods (DROP-Grad, DROP-Best-Ada, and DROP-Grad-Ada), Rank is more stable and yields better performance compared with the other two decomposition rules.

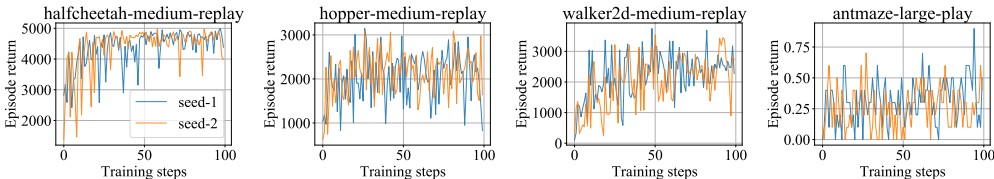

Figure 8: Learning curves of DROP, where the x-axis denotes the training steps (k), the y-axis denotes the evaluation return (using DROP-Best embedding inference rule). We only show two seeds for legibility.

To answer the above question, we compare DROP (using the Rank decomposition rule) with filtered behavior cloning (F-BC), where the latter (F-BC) performs behavior cloning after filtering for trajectories with high returns. We provide the comparison results in Table 5. We can find that in AntMaze tasks, the overall performance of DROP is higher than that of F-BC. For the MuJoCo-Gym suite, DROP-based methods outperform F-BC on these offline tasks that contain plenty of sub-optimal trajectories, including the random, medium, and medium-replay domains. This result indicates that DROP can leverage embedding inference (extrapolation) to find a better policy beyond all the behavior policies in sub-tasks, which is more effective than simply performing imitation learning on a subset of the dataset.

## A.3 Online Fine-tuning

**Online fine-tuning (checkpoint-level).** In Figure 8, we show the learning curves of DROP-Best on four DR4L tasks. We can find that DROP exhibits a high variance (in performance) across training

Table 5: Comparison between our DROP (using the Rank decomposition rule) and filtered behavior cloning (F-BC) on D4RL AntMaze and MuJoCo suites (*-v2). We take the baseline results of BC and F-BC from Emmons et al. [14], where F-BC is trained over the top $10\%$ trajectories, ordered by the returns. Our DROP results are computed over 5 seeds and 10 episodes for each seed.

| Tasks | | BC | F-BC | DROP-Grad | DROP-Best-Ada | DROP-Grad-Ada |
|---|---|---|---|---|---|---|
| antmaze-umaze | | 54.6 | 60 | $72 \pm 17.2$ | $78 \pm 11.7$ | $80 \pm 12.6$ |
| antmaze-umaze-diverse | | 45.6 | 46.5 | $48 \pm 22.3$ | $62 \pm 16$ | $66 \pm 12$ |
| antmaze-medium-play | | 0 | 42.1 | $24 \pm 10.2$ | $34 \pm 12$ | $30 \pm 21$ |
| antmaze-medium-diverse | | 0 | 37.2 | $20 \pm 19$ | $24 \pm 12$ | $30 \pm 16.7$ |
| antmaze-large-play- | | 0 | 28 | $24 \pm 8$ | $36 \pm 17.4$ | $42 \pm 17.2$ |
| antmaze-large-diverse | | 0 | 34.3 | $14 \pm 8$ | $20 \pm 14.1$ | $26 \pm 13.6$ |
| halfcheetah | random | 2.3 | 2 | $2.3 \pm 0$ | $2.3 \pm 0$ | $2.3 \pm 0$ |
| hopper | random | 4.8 | 4.1 | $5.1 \pm 0.8$ | $5.4 \pm 0.7$ | $5.5 \pm 0.6$ |
| walker2d | random | 1.7 | 1.7 | $2.8 \pm 1.7$ | $3 \pm 1.6$ | $3 \pm 1.8$ |
| halfcheetah | medium | 42.6 | 42.5 | $42.4 \pm 0.7$ | $42.9 \pm 0.4$ | $43.1 \pm 0.4$ |
| hopper | medium | 52.9 | 56.9 | $57.5 \pm 6.4$ | $60.3 \pm 6.1$ | $59.5 \pm 5.1$ |
| walker2d | medium | 75.3 | 75 | $76.5 \pm 2.4$ | $75.8 \pm 3$ | $79.1 \pm 1.4$ |
| halfcheetah | medium-replay | 36.6 | 40.6 | $39.5 \pm 1$ | $40.4 \pm 0.8$ | $40.3 \pm 1.2$ |
| hopper | medium-replay | 18.1 | 75.9 | $48 \pm 17.7$ | $83.4 \pm 6.5$ | $87.4 \pm 2.1$ |
| walker2d | medium-replay | 26 | 62.5 | $37.4 \pm 13.5$ | $60.9 \pm 7.4$ | $61.9 \pm 2.3$ |

steps[9], which means the performance of the agent may be dependent on the specific stopping point chosen for evaluation (such instability also exists in prior offline RL methods, Fujimoto and Gu [19]).

To choose a suitable stopping checkpoint over which we perform the DROP inference (DROP-Grad, DROP-Best-Ada and DROP-Grad-Ada), we propose to conduct *checkpoint-level* online fine-tuning (see Algorithm 3 in Section B for more details): we evaluate each of the latest $T$ checkpoint models and choose the best one that leads to the highest episode return.

In Figure 9, we show the total normalized returns across all the tasks in each suite (including Maze2d, AntMaze, and MuJoCo-Gym). We can find that in most tasks, fine-tuning (FT) can guarantee performance improvement. However, we also find such fine-tuning causes negative impacts on performance in AntMaze(*-v0) suite. The main reason is that, in this checkpoint-level fine-tuning, we choose the "suitable" checkpoint model using the DROP-Best embedding inference rule, while we adopt the other three embedding inference rules (DROP-Grad, DROP-Best-Ada and DROP-Grad-Ada) at the test time. Such a finding also implies that the success of DROP's test-time adaptation is not entirely dependent on the best embedding across sub-tasks [10] (*i.e.*, the best embedding $\mathbf{z}_0^*(\mathbf{s}_0)$ in DROP-Best), but requires switching between some "suboptimal" embeddings (using DROP-Best-Ada) or extrapolating new embeddings (using DROP-Grad-Ada).

**Online fine-tuning (embedding-level).** Beyond the above checkpoint-level fine-tuning procedure, we can also conduct *embedding-level* online fine-tuning: we aim to choose a suitable gradient update step for the gradient-based embedding inference rules (including DROP-Grad and DROP-Grad-Ada). Similar to the checkpoint-level fine-tuning, we first conduct the test-time adaptation procedure (DROP-Grad and DROP-Grad-Ada) over a set of gradient update steps, and then choose the best step that leads to the highest episode return (see Algorithm 4 in Section B for more details).

---

[9] In view of such instability, we evaluate our methods over multiple checkpoints for each seed, instead of choosing the final checkpoint models during the training loop (see the detailed evaluation protocol in Appendix B).

[10] Conversely, if the performance of DROP depends on the best embedding across sub-tasks (*i.e.*, $\mathbf{z}_0^*(\mathbf{s}_0)$ in DROP-Best), then the checkpoint model we choose by fine-tuning with DROP-Best should enable a consistent performance improvement for rules that perform embedding inference with DROP-Best-Ada and DROP-Grad-Ada. However, we find a performance drop in AntMaze(*-v0) suite, which means there is no explicit dependency between the best embedding $\mathbf{z}_0^*(\mathbf{s}_0)$ and the inferred embedding using the adaptive inference rules.

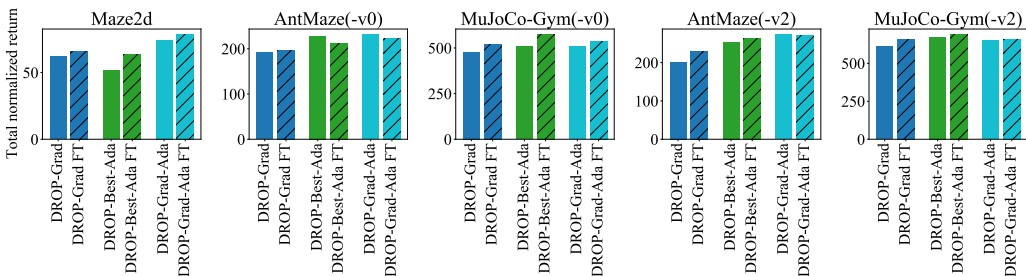

Figure 9: Total normalized returns across all the tasks in Maze2d, AntMaze, and MuJoCo-Gym suites.

Table 6: Online fine-tuning results (initial performance → performance after online fine-tuning). The baseline results of AWAC, CQL, and IQL are taken from Kostrikov et al. [31], where they run 1M online steps to fine-tune the learned policy. For our DROP method (DROP-Grad and DROP-Grad-Ada), we run 0.3M ($= 6_{\text{checkpoint}} \times 50_{K_{\max}} \times 1000_{\text{steps per episode}}$) online steps to fine-tune (embedding-level) the policy, *i.e.*, aiming to find the optimal gradient ascent step that is used to infer the contextual embedding $\mathbf{z}^*(\mathbf{s}_0)$ or $\mathbf{z}^*(\mathbf{s}_t)$ for $\pi^*(\mathbf{a}_t|\mathbf{s}_t) := \beta(\mathbf{a}_t|\mathbf{s}_t, \cdot)$ (see Algorithm 4 for the details). Moreover, for medium-* and large-* tasks, we conduct additional parametric-level fine-tuning, with 0.7M online steps to update the policy's parameters.

| Task (*-v0) | CQL | IQL | DROP-Grad | DROP-Grad-Ada | |
|---|---|---|---|---|---|
| umaze | 70.1 → 99.4 | 86.7 → 96 | 70 → 96 ± 1.2 | 76 → 98 ± 0 | |
| umaze-diverse | 31.1 → 99.4 | 75 → 84 | 54 → 88 ± 8 | 66 → 94 ± 4.9 | |
| medium-play | 0 23 → 0 | 72 → 95 | 20 → 56 ± 8.9 | 30 → 50 ± 6.3 | → 94 ± 2.9 |
| medium-diverse | 23 → 32.3 | 68.3 → 92 | 12 → 44 ± 4.9 | 22 → 38 ± 4.9 | → 96 ± 0.8 |
| large-play | 1 → 0 | 25.5 → 46 | 16 → 38 ± 8.9 | 16 → 40 ± 6.3 | → 53 ± 1.3 |
| large-diverse | 1 → 0 | 42.6 → 60.7 | 20 → 40 ± 13.6 | 22 → 46 ± 10.2 | → 58 ± 4.5 |
| | →⏟1M | →⏟1M | →⏟0.3M | →⏟0.3M | →⏟0.7M |

In Table 6, we compare our DROP (DROP-Grad and DROP-Grad-Ada) to three offline RL methods (AWAC [53], CQL [35] and IQL [31]), reporting the initial performance and the performance after online fine-tuning. We can find that the embedding-level fine-tuning (0.3M) enables a significant improvement in performance. The fine-tuned DROP-Grad-Ada (0.3M) outperforms the AWAC and CQL counterparts in most tasks, even though we take fewer rollout steps to conduct the online fine-tuning (baselines take 1M online rollout steps, while DROP-based fine-tuning takes 0.3M steps). However, there is still a big gap between the fine-tuned IQL and the embedding-level fine-tuned DROP (0.3M). Considering that there remain 0.7M online steps in the comparison, we further conduct "parametric-level" fine-tuning (updating the parameters of the policy network) for our DROP-Grad-Ada on medium-* and large-* tasks, we can find which achieves competitive fine-tuning performance even compared with IQL.

## A.4  DROP + CVAE Implementation

**CVAE-based embedding learning.** Similar to LAPO [11] and PLAS [74], we adopt the conditional variational auto-encoder (CVAE) to model offline data. Specifically, we learn the contextual policy and behavior embedding:

$$\beta(\mathbf{a}|\mathbf{s},\mathbf{z}), \phi(\mathbf{z}|\mathbf{s}) \leftarrow \arg\max_{\beta,\phi} \mathbb{E}_{(\mathbf{s},\mathbf{a})\sim\mathcal{D}}\mathbb{E}_{(\mathbf{z})\sim\phi(\mathbf{z}|\mathbf{s})}\big[\log\beta(\mathbf{a}|\mathbf{s},\mathbf{z})\big] - \text{KL}(\phi(\mathbf{z}|\mathbf{s})\|p(\mathbf{z})). \quad (10)$$

Then, we learn the score model $f$ with the TD-error and the conservative regularization:

$$f \leftarrow \arg\min_{f} \mathbb{E}_{(\mathbf{s},\mathbf{a},\mathbf{s}',\mathbf{a}')\sim\mathcal{D}}\Big[\big(R(\mathbf{s},\mathbf{a}) + \gamma\bar{f}(\mathbf{s}',\mathbf{a}',\phi(\mathbf{z}|\mathbf{s})) - f(\mathbf{s},\mathbf{a},\phi(\mathbf{z}|\mathbf{s}))\big)^2\Big], \quad (11)$$

$$\text{s.t. } \mathbb{E}_{\mathbf{s}\sim\mathcal{D},\mathbf{z}\sim\mu(\mathbf{z}),\mathbf{a}\sim\beta(\mathbf{a}|\mathbf{s},\mathbf{z})}\left[f(\mathbf{s},\mathbf{a},\mathbf{z})\right] - \mathbb{E}_{\mathbf{s}\sim\mathcal{D},\mathbf{z}\sim\phi(\mathbf{z}|\mathbf{s}),\mathbf{a}\sim\beta(\mathbf{a}|\mathbf{s},\mathbf{z})}\left[f(\mathbf{s},\mathbf{a},\mathbf{z})\right] \leq \eta,$$

where $\bar{f}$ denotes a target network and $\mu(\mathbf{z})$ denotes the uniform distribution over the $\mathcal{Z}$-space.

In testing, we also dynamically adapt the outer-level optimization, setting policy inference with $\pi^*(\mathbf{a}|\mathbf{s}) = \beta(\mathbf{a}|\mathbf{s}, \mathbf{z}^*(\mathbf{s}))$, where $\mathbf{z}^*(\mathbf{s}) = \arg\max_z f(\mathbf{s}, \beta(\mathbf{a}|\mathbf{s}, \mathbf{z}), \mathbf{z})$.

### A.5  Ablation Study

Note that our embedding inference depends on the learned score model $f$. Without proper regularization, such inference will lead to out-of-distribution embeddings that are erroneously high-scored (Q2). Here we conduct an ablation study to examine the impact of the conservative regularization used for learning the score model.

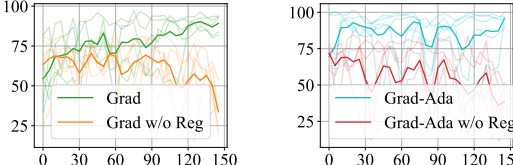

Figure 10: Ablation on the conservative regularization. The y-axis represents the normalize return, and the x-axis represents the number of gradient-ascent steps used for embedding inference at deployment. We plot each random seed as a transparent line; the solid line corresponds to the average across 5 seeds.

In Figure 10, we compare DROP-Grad and DROP-Grad-Ada to their naive implementation (*w/o Reg*) that ablates the regularization on halfcheetah-medium-expert. We can find that removing the conservative regularization leads to unstable performance when changing the update steps of gradient-based optimization. However, we empirically find that in some tasks such a naive implementation (*w/o Reg*) does not necessarily bring unstable inference (Appendix C). Although improper gradient update step leads to faraway embeddings, to some extent, embedding-conditioned behavior policy can correct such deviation.

## B  Implementation Details

For the practical implementation of DROP, we parameterize the task embedding function $\phi(\mathbf{z}|n)$, the contextual behavior policy $\beta(\mathbf{a}|\mathbf{s}, \mathbf{z})$, and the score model $f(\mathbf{s}, \mathbf{a}, \mathbf{z})$ with neural networks. For Equation 8 in the main paper, we construct a Lagrangian and solve the optimization through primal-dual gradient descent. For the choice of $\mu(\mathbf{z})$, we simply set $\mu(\mathbf{z})$ to be the uniform distribution over the $\mathcal{Z}$-space and empirically find that such uniform sampling can effectively avoid the out-of-distribution extrapolation at inference.

**Lagrangian relaxation.**  To optimize the constrained objective in Equation 8 in the main paper, we construct a Lagrangian and solve the optimization through primal-dual gradient descent,

$$\min_f \max_{\lambda > 0} \quad \mathbb{E}_{\mathcal{D}_n \sim \mathcal{D}_{[N]}} \mathbb{E}_{(\mathbf{s}, \mathbf{a}, \mathbf{s}', \mathbf{a}') \sim \mathcal{D}_n} \left[ \left( R(\mathbf{s}, \mathbf{a}) + \gamma \bar{f}(\mathbf{s}', \mathbf{a}', \phi(\mathbf{z}|n)) - f(\mathbf{s}, \mathbf{a}, \phi(\mathbf{z}|n)) \right)^2 \right] +$$
$$\lambda \left( \mathbb{E}_{n, \mu(\mathbf{z})} \mathbb{E}_{\mathbf{s} \sim \mathcal{D}_n, \mathbf{a} \sim \beta(\mathbf{a}|\mathbf{s}, \mathbf{z})} \left[ f(\mathbf{s}, \mathbf{a}, \mathbf{z}) \right] - \mathbb{E}_{n, \phi(\mathbf{z}|n)} \mathbb{E}_{\mathbf{s} \sim \mathcal{D}_n, \mathbf{a} \sim \beta(\mathbf{a}|\mathbf{s}, \mathbf{z})} \left[ f(\mathbf{s}, \mathbf{a}, \mathbf{z}) \right] - \eta \right).$$

This unconstrained objective implies that if the expected difference in scores of out-of-distribution embeddings and in-distribution embeddings is less than a threshold $\eta$, $\lambda$ is going to be adjusted to $0$, on the contrary, $\lambda$ is likely to take a larger value, used to punish the over-estimated value function. This objective encourages that out-of-distribution embeddings score lower than in-distribution embeddings, thus performing embedding inference will not lead to these out-of-distribution embeddings that are falsely and over-optimistically scored by the learned score model.

In our experiments, we tried five different values for the Lagrange threshold $\eta$ (1.0, 2.0, 3.0, 4.0, and 5.0). We did not observe a significant difference in performance across these values. Therefore, we simply set $\eta = 2.0$.

**Hyper-parameters.**  In Table 7, we provide the hyper-parameters of the task embedding $\phi(\mathbf{z}|\mathbf{s})$, the contextual behavior policy $\beta(\mathbf{a}|\mathbf{s}, \mathbf{z})$, and the score function $f(\mathbf{s}, \mathbf{a}, \mathbf{z})$. For the gradient ascent update steps (used for embedding inference), we set $K = 100$ for all the embedding inference rules in experiments.

In Table 9, we provide the number of sub-tasks, the number of trajectories in each sub-task, and the dimension of the embedding for each sub-task (behavior policy). The selection of hyperparameter N is based on two evaluation metrics: (1) the fitting loss of the decomposed behavioral policies to the offline data, and (2) the testing performance of DROP. Specifically,

Table 7: Hyper-parameters of the task embedding function $\phi(\mathbf{z}|\mathbf{s})$, the contextual behavior policy $\beta(\mathbf{a}|\mathbf{s}, \mathbf{z})$, and the score function $f(\mathbf{s}, \mathbf{a}, \mathbf{z})$. The task embedding function $\phi(\mathbf{z}|\mathbf{s})$: $\mathbf{z} \leftarrow \mathrm{Enc\_0}(n)$. The contextual behavior policy $\beta(\mathbf{a}|\mathbf{s}, \mathbf{z})$: $\mathbf{a} \leftarrow \mathrm{Enc\_2}(\mathbf{z}, \mathrm{Enc\_1}(\mathbf{s}))$. The score function $f(\mathbf{s}, \mathbf{a}, \mathbf{z})$: $f \leftarrow \mathrm{Enc\_4}(\mathbf{z}, \mathrm{Enc\_3}(\mathbf{s}, \mathbf{a}))$.

|  | Enc_0 | Enc_1 | Enc_2 | Enc_3 | Enc_4 |
|---|---|---|---|---|---|
| Optimizer | Adam | Adam | Adam | Adam | Adam |
| Hidden layer | 2 | 2 | 3 | 2 | 3 |
| Hidden dim | 512 | 512 | 512 | 512 | 512 |
| Activation function | ReLU | ReLU | ReLU | ReLU | ReLU |
| Learning rate | 1.00E-03 | 1.00E-03 | 1.00E-03 | 1.00E-03 | 1.00E-03 |
| Mini-batch size | 1024 | 1024 | 1024 | 1024 | 1024 |

- (Step1) Over a hyperparameter (the number of sub-tasks) set, we conduct the hyperparameter search using the fitting loss of behavior policies, then we choose/filter the four best hyperparameters;
- (Step2) We follow the normal practice of hyperparameter selection and tune the four hypermeters selected in Step1 by interacting with the simulator to estimate the performance of DROP under each hyperparameter setting.

We provide the hyperparameter sets in Table 8. In Step2, we tune the (filtered) hyperparameters using 1 seed, then evaluate the best hyperparameter by training on an additional 4 seeds and finally report the results on the 5 total seeds (see next "evaluation protocol"). In Antmaze domain, a single fixed N works well for many tasks;

Table 8: Hyperparameter (the number of sub-tasks) set.

| tasks | the number of sub-tasks |
|---|---|
| Antmaze | 500 (v0), 150 (v2) |
| Gym-mujoco | 10, 20, 50, 100, 200, 500, 800, 1000 |
| Adroit | 10, 20, 50, 100, 200, 500, 800, 1000 |

while in Gym-mujoco and Adroit domains, we did not find a fixed N that provides good results for all tasks in the corresponding domain in D4RL, thus we use the above hyperparameter selection rules (Step1 and Step2) to choose the number.

**Baseline details.** For the comparison of our method to prior iterative offline RL methods, we consider the v0 versions of the datasets in D4RL[11]. We take the baseline results of BEAR, BCQ, CQL, and BRAC-p from the D4RL paper [18], and take the results of TD3+BC from their origin paper [19]. For the comparison of our method to prior non-iterative offline RL method, we use the v2 versions of the dataset in D4RL. All the baseline results of behavior cloning (BC), Decision Transform (DT), RvS-R, and Onestep are taken from Emmons et al. [14]. In our implementation of COMs, we take the parameters (neural network weights) of behavior policies as the design input for the score model; and during testing, we conduct parameters inference (outer-level optimization) with 200 steps gradient ascent over the learned score function, then the rollout policy is initialized with the inferred parameters. For the specific architecture, we instantiate the policy network with $\dim(\mathcal{S})$ input units, two layers with 64 hidden units, and a final output layer with $\dim(\mathcal{A})$.

**Evaluation protocol.** We evaluate our results over 5 seeds. For each seed, instead of taking the final checkpoint model produced by a training loop, we take the last $T$ ($T = 6$ in our experiments) checkpoint models, and evaluate them over 10 episodes for each checkpoint. That is to say, we report the average of the evaluation scores over $5_{\text{seed}} \times 6_{\text{checkpoint}} \times 10_{\text{episode}}$ rollouts.

*Online fine-tuning (checkpoint-level):* Instead of re-training the learned (final) policy with online rollouts, we fine-tune our policy with enumerated trial-and-error over the last $T$ checkpoint models (Algorithm 3). Specifically, for each seed, we run the last $T$ checkpoint models in the environment over one episode for each checkpoint. The checkpoint model which achieves the maximum episode return is returned. In essence, this fine-tuning procedure imitates the online RL evaluation protocol: if the current policy is unsatisfactory, we can use checkpoints of previous iterations of the policy.

---

[11]We noticed that Maze2D-v0 in the D4RL dataset (https://rail.eecs.berkeley.edu/datasets/) is not available, so we used v1 version instead in our experiment. For simplicity, we still use v0 in the paper exposition.

---

**Algorithm 3** DROP: Online fine-tuning (checkpoint-level)

---

**Require:** Env, last $T$ checkpoint models: $\beta_t(\mathbf{a}|\mathbf{s}, \mathbf{z})$ and $f_t(\mathbf{s}, \mathbf{a}, \mathbf{z})$ ($t = 1, \cdots, T$).

1: $R_{\text{MAX}} = -\infty$.
2: $\beta_{\text{best}} \leftarrow$ None.
3: $f_{\text{best}} \leftarrow$ None.
4: **while** $t = 1, \cdots, T$ **do**
5:     $\mathbf{s}_0 =$ Env.Reset().
6:     $\mathbf{z}_0^*(\mathbf{s}_0) \leftarrow$ Conduct embedding inference with DROP-Best.
7:     Return $\leftarrow$ Evaluate $\beta_t$ and $f_t$ on Env, setting $\pi^*(\mathbf{a}|\mathbf{s}) = \beta(\mathbf{a}|\mathbf{s}, \mathbf{z}_0^*(\mathbf{s}_0))$.
8:     **if** $R_{\text{MAX}} <$ Return **then**
9:         Update the best checkpoint models: $\beta_{\text{best}} \leftarrow \beta_t$, $f_{\text{best}} \leftarrow f_t$.
10:        Update the optimal return: $R_{\text{MAX}} \leftarrow$ Return.
11:     **end if**
12: **end while**

**Return:** $\beta_{\text{best}}$ and $f_{\text{best}}$.

---

**Algorithm 4** DROP: Online fine-tuning (embedding-level)

---

**Require:** Env, last $T$ checkpoint models: $\beta_t(\mathbf{a}|\mathbf{s}, \mathbf{z})$ and $f_t(\mathbf{s}, \mathbf{a}, \mathbf{z})$ ($t = 1, \cdots, T$).

1: $R_{\text{MAX}} = -\infty$.
2: $\beta_{\text{best}} \leftarrow$ None.
3: $f_{\text{best}} \leftarrow$ None.
4: $k_{\text{best}} \leftarrow 0$.
5: **while** $t = 1, \cdots, T$ **do**
6:     **while** $k = 1, \cdots, K_{\max}$ **do**
7:         $\mathbf{s}_0 =$ Env.Reset().
          # Conduct embedding inference with DROP-Grad *or* DROP-Grad-Ada
8:         Return $\leftarrow$ Evaluate $\beta_t$ and $f_t$ on Env, setting $\pi^*(\mathbf{a}|\mathbf{s}) = \beta(\mathbf{a}|\mathbf{s}, \mathbf{z}^*(\mathbf{s}_0))$ *or* $\beta(\mathbf{a}|\mathbf{s}, \mathbf{z}^*(\mathbf{s}))$, where we conduct $k$ gradient ascent steps to obtain $\mathbf{z}^*(\mathbf{s}_0)$ *or* $\mathbf{z}^*(\mathbf{s})$.
9:         **if** $R_{\text{MAX}} <$ Return **then**
10:           Update the best checkpoint models: $\beta_{\text{best}} \leftarrow \beta_t$, $f_{\text{best}} \leftarrow f_t$.
11:           Update the best gradient update step: $k_{\text{best}} \leftarrow k$.
12:           Update the optimal return: $R_{\text{MAX}} \leftarrow$ Return.
13:         **end if**
14:     **end while**
15: **end while**

**Return:** $\beta_{\text{best}}$, $f_{\text{best}}$ and $k_{\text{best}}$.

---

*Online fine-tuning (embedding-level):* The embedding-level fine-tuning aims to find a suitable gradient ascent step that is used to conduct the embedding inference in DROP-Grad *or* DROP-Grad-Ada. Thus, we enumerate a list of gradient update steps and pick the best update step (according to the episode returns).

**Codebase.** Our code is based on d3rlpy: https://github.com/takuseno/d3rlpy. We provide our source code in the supplementary material.

**Computational resources.** The experiments were run on a computational cluster with 22x GeForce RTX 2080 Ti, and 4x NVIDIA Tesla V100 32GB for 20 days.

## C Additional Results

**Comparison with iterative offline RL baselines.** Here, we compare the performance of DROP (Grad, Best-Ada, and Grad-Ada ) to iterative offline RL baselines (BEAR [33], BCQ [20], CQL [35], BRAC-p [67], and TD3+BC [19]) that perform iterative bi-level offline RL paradigm with (explicit or implicit) value/policy regularization in inner-level. In Table 10, we present the results for AntMaze, Gym-MuJoCo, and Adroit suites in standard D4RL benchmark (*-v0), where we can find that DROP-Grad-Ada performs comparably or surpasses prior iterative bi-level works on most tasks: outperforming (or comparing) these policy regularized methods (BRAC-p and TD3+BC) on 25 out of 33 tasks and outperforming (or comparing) these value regularized algorithms (BEAR, BCQ, and CQL) on 19 out of 33 tasks.

Table 9: The number ($N$) of sub-tasks, the number ($M$) of trajectories in each sub-task, and the dimension ($\dim(\mathbf{z})$) of the embedding for each sub-task.

| Domain | Task Name | Parameters (*-v0) | | | Parameters (*-v2) | | |
|---|---|---|---|---|---|---|---|
| | | $N$ | $M$ | $\dim(\mathbf{z})$ | $N$ | $M$ | $\dim(\mathbf{z})$ |
| Maze 2D | umaze | 500 | 5 | 5 | | | |
| | medium | 150 | 50 | 5 | | | |
| | large | 100 | 15 | 5 | | | |
| Antmaze | umaze | 500 | 50 | 5 | 150 | 50 | 5 |
| | umaze-diverse | 500 | 50 | 5 | 150 | 50 | 5 |
| | Medium-play | 500 | 50 | 5 | 150 | 50 | 5 |
| | Medium-diverse | 500 | 50 | 5 | 150 | 50 | 5 |
| | Large-play | 500 | 50 | 5 | 150 | 50 | 5 |
| | Large-diverse | 500 | 50 | 5 | 150 | 50 | 5 |
| halfcheetah | random | 1000 | 1 | 5 | 1000 | 1 | 5 |
| | medium | 100 | 2 | 5 | 100 | 2 | 5 |
| | medium-expert | 1000 | 1 | 5 | 1000 | 1 | 5 |
| | medium-replay | 50 | 10 | 5 | 50 | 10 | 5 |
| hopper | random | 100 | 2 | 5 | 100 | 2 | 5 |
| | medium | 100 | 5 | 5 | 100 | 5 | 5 |
| | medium-expert | 100 | 2 | 5 | 100 | 2 | 5 |
| | medium-replay | 50 | 5 | 5 | 10 | 30 | 5 |
| walker2d | random | 500 | 2 | 5 | 500 | 2 | 5 |
| | medium | 50 | 5 | 5 | 50 | 5 | 5 |
| | medium-expert | 50 | 5 | 5 | 50 | 5 | 5 |
| | medium-replay | 1000 | 5 | 5 | 10 | 50 | 5 |
| door | cloned | 1000 | 2 | 5 | | | |
| | expert | 500 | 5 | 5 | | | |
| | human | 50 | 3 | 5 | | | |
| hammer | cloned | 1000 | 1 | 5 | | | |
| | expert | 500 | 5 | 5 | | | |
| | human | 20 | 3 | 5 | | | |
| pen | cloned | 500 | 5 | 5 | | | |
| | expert | 500 | 5 | 5 | | | |
| | human | 50 | 5 | 5 | | | |
| relocate | cloned | 500 | 5 | 5 | | | |
| | expert | 500 | 5 | 5 | | | |
| | human | 50 | 4 | 5 | | | |

**Ablation studies.** In Figure 11, we provide more results for the ablation of the conservative regularization term in Equation 8 in the main paper. We can find that for the halfcheetah-medium and hopper-medium tasks, the performance of DROP-Grad-Ada w/o Reg depends on the choice of the gradient update steps, showing that too small or too large number of gradient update step deteriorates the performance. Such a result is also consistent with COMs [62], which also observes the sensitivity of naive gradient update (*i.e.*, *w/o Reg*) to the number of update steps used for design input inference. By comparison, the conservative score model learned with DROP-Grad-Ada exhibits more stable and robust performance to the gradient update steps.

Further, we also find that in walker2d-medium and walker2d-medium-expert tasks, the naive gradient update (*w/o Reg*) does not affect performance significantly across a wide range of gradient update steps. The main reason is that although the excessive gradient updates lead to faraway embeddings, conditioned on the inferred embeddings, the learned contextual behavior policy can safeguard against the embeddings distribution shift. Compared to prior model-based optimization that conducts direct gradient optimization (inference) over the design input itself, such "self-safeguard" is a special merit in the offline RL domain as long as we reframe the offline RL problem as one of model-based optimization and conduct inference over the embedding space. Thus, we encourage the research community to pursue further into this model-based optimization view for the offline RL problem.

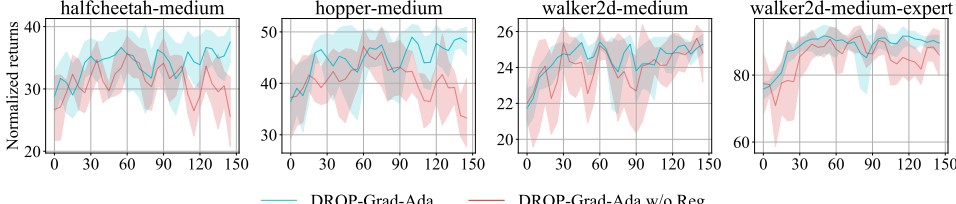

Figure 11: The performance comparison of DROP-Grad-Ada and DROP-Grad-Ada w/o Reg, where we ablate the conservative regularization for the *w/o Reg* implementation. The y-axis denotes the normalized return, the x-axis denotes the number of gradient-ascent steps used for embedding inference at deployment.

Table 10: Comparison of our method to prior offline methods that perform iterative (regularized) RL paradigm on D4RL. We take the baseline results of BEAR, BCQ, CQL, and BRAC-p from Fu et al. [18], and the results of TD3-BC from Fujimoto and Gu [19]. For all results of our method (DROP), we average the normalized returns across 5 seeds; for each seed, we run 10 evaluation episodes. For proper comparison, we use ▲ and ▲ to denote DROP (*-Ada) achieves *comparable or better* performance compared with value and policy regularized offline RL methods respectively.

| | Task Name | Value Reg. | | | Policy Reg. | | DROP- | | |
|---|---|---|---|---|---|---|---|---|---|
| | | BEAR | BCQ | CQL | BRAC-p | TD3+BC | Grad | Best-Ada | Grad-Ada |
| antmaze | umaze | 73.0 | 78.9 | 74.0 | 50.0 | – | 72.0 | 78.0▲▲ | 80.0▲▲ |
| | umaze-diverse | 61.0 | 55.0 | 84.0 | 40.0 | – | 48.0 | 62.0  ▲ | 66.0  ▲ |
| | medium-play | 0.0 | 0.0 | 61.2 | 0.0 | – | 24.0 | 34.0  ▲ | 30.0  ▲ |
| | medium-diverse | 8.0 | 0.0 | 53.7 | 0.0 | – | 20.0 | 24.0  ▲ | 30.0  ▲ |
| | large-play | 0.0 | 6.7 | 15.8 | 0.0 | – | 24.0 | 36.0▲▲ | 42.0▲▲ |
| | large-diverse | 0.0 | 2.2 | 14.9 | 0.0 | – | 14.0 | 20.0▲▲ | 26.0▲▲ |
| halfcheetah | random | 25.1 | 2.2 | 35.4 | 24.1 | 10.2 | 2.3 | 2.3 | 2.3 |
| | medium | 41.7 | 40.7 | 44.4 | 43.8 | 42.8 | 42.4 | 42.9▲▲ | 43.1▲▲ |
| | medium-expert | 53.4 | 64.7 | 62.4 | 44.2 | 97.9 | 86.6 | 88.5▲ | 88.9▲ |
| | medium-replay | 38.6 | 38.2 | 46.2 | 45.4 | 43.3 | 39.5 | 40.4 | 40.3 |
| hopper | random | 11.4 | 10.6 | 10.8 | 11.0 | 11.0 | 5.1 | 5.4 | 5.5 |
| | medium | 52.1 | 54.5 | 58.0 | 32.7 | 99.5 | 57.5 | 60.3▲ | 59.5▲ |
| | medium-expert | 96.3 | 110.9 | 98.7 | 1.9 | 112.2 | 103.5 | 102.5 | 105.9▲ |
| | medium-replay | 33.7 | 33.1 | 48.6 | 0.6 | 31.4 | 48.0 | 83.4▲▲ | 87.4▲▲ |
| walker2d | random | 7.3 | 4.9 | 7.0 | -0.2 | 1.4 | 2.8 | 3.0  ▲ | 3.0  ▲ |
| | medium | 59.1 | 53.1 | 79.2 | 77.5 | 79.7 | 76.5 | 75.8▲▲ | 79.1▲▲ |
| | medium-expert | 40.1 | 57.5 | 111.0 | 76.9 | 101.1 | 107.5 | 106.8▲▲ | 106.9▲▲ |
| | medium-replay | 19.2 | 15.0 | 26.7 | -0.3 | 25.2 | 37.4 | 60.9▲▲ | 61.9▲▲ |
| door | cloned | -0.1 | 0.0 | 0.4 | -0.1 | – | 0.5 | 2.5 | 2.7▲▲ |
| | expert | 103.4 | 99.0 | 101.5 | -0.3 | – | 98.6 | 102.2▲▲ | 102.6▲▲ |
| | human | -0.3 | 0.0 | 9.9 | -0.3 | – | 3.3 | 1.9  ▲ | 3.0  ▲ |
| hammer | cloned | 0.3 | 0.4 | 2.1 | 0.3 | – | 0.3 | 0.3  ▲ | 0.3  ▲ |
| | expert | 127.3 | 107.2 | 86.7 | 0.3 | – | 65.7 | 73.3  ▲ | 77.7  ▲ |
| | human | 0.3 | 0.5 | 4.4 | 0.3 | – | 1.1 | 0.3 | 2.1  ▲ |
| pen | cloned | 26.5 | 44.0 | 39.2 | 1.6 | – | 76.7 | 77.1▲▲ | 82.4▲▲ |
| | expert | 105.9 | 114.9 | 107.0 | -3.5 | – | 113.1 | 118.6▲▲ | 116.7▲▲ |
| | human | -1.0 | 68.9 | 37.5 | 8.1 | – | 71.1 | 85.2▲▲ | 81.5▲▲ |
| relocate | cloned | -0.3 | -0.3 | -0.1 | -0.3 | – | 0.1 | 0.5▲▲ | 0.2▲▲ |
| | expert | 98.6 | 41.6 | 95.0 | -0.3 | – | 2.5 | 6.2  ▲ | 5.4  ▲ |
| | human | -0.3 | -0.1 | 0.2 | -0.3 | – | 0.0 | 0.0 | 0.0 |

