# OpenReview forum: "Design from Policies: Conservative Test-Time Adaptation for Offline Policy Optimization"
_NeurIPS.cc/2023/Conference — NeurIPS 2023 poster_

### Official Review · Reviewer_SsTF · 2023-06-19

**Soundness:** 3 good
**Presentation:** 2 fair
**Contribution:** 3 good
**Rating:** 5
**Confidence:** 4

**Summary:**

This paper proposes an iterative bi-level offline RL algorithm that separates an "inner level" and "outer level" optimization for reinforcement learning. A novel variant of model-based optimization (including task decomposition and task embedding) is proposed to enable flexible test-time adaptation. Experiments on popular d4rl benchmarks show state-of-the-art performances over other recent baselines.

**Strengths:**

This paper proposes an approach that incorporates model-based optimization into offline rl with test-time adaptation. Each component in the methodology is well justified. This reformulation of offline rl as model-based optimization seems novel and can be of value to the community.

Adequate comparisons have been made to prior iterative and non-iterative offline rl methods, and the differences between the proposed method and related works are well justified.

Systematic evaluations on popular D4RL benchmarks are included, and statistically significant improvement over other state-of-the-art methods is observed.

The weakness of increased computation cost is discussed.

**Weaknesses:**

The writing of the paper (especially that of the abstract and introduction) is hard to follow. Specifically, the terms "inner level" and "outer level" are very confusing. Using other terms such as "value estimation" and "policy extraction" or the like will make reading much easier. The three questions Q1, Q2, and Q3 raised in the abstract and introduction are also very vague and do not make sense until I have carefully read the methodology and experiments. In my opinion, the main contribution of this paper is reformulating offline rl as a model-based optimization problem and it makes flexible adaptation to offline states possible.

I am missing a comparison to related works in offline rl with test time adaptation. In particular, a comparison with [x] should be discussed.

In Figure 5, is it possible to also include the inference time of other baselines (COMs, RvS-R, etc)? How do Onestep and DROP compare when they use the same amount of computation?

As discussed in the limitations section, the number of subtasks can be important for this method. However, in offline rl when online data is difficult to collect, we are not allowed to tune hyperparameters using online interactions [y]. The procedure of determining hyperparameters should be included in the paper.


[x] Offline RL Policies Should be Trained to be Adaptive (https://arxiv.org/abs/2207.02200)

[y] Conservative Objective Models for Effective Offline Model-Based Optimization (https://arxiv.org/pdf/2107.06882)

**Questions:**

In Table 2, which variant is the listed DROP? DROP Best, Grad, Best-Ada, or Grad-Ada?

When comparing DROP with LAPO in Table 2, why are the results of LAPO on u, um-p, and ul-p missing? How is the mean calculated with missing entry?

In Figure 5, the inference time of Diffuser is reported, is it possible to also report its performance in Table 2 or Figure 4?

In line 226-228, "Different from this iterative paradigm, DROP only evaluates values of behavior policies in the inner-level optimization, avoiding the potential overestimation for values of new learning policies and eliminating the error propagation between the two levels." I do not quite understand why the iterative process can cause error propagation for potential overestimation of values, if you consider the iterative process as a min max optimization process? And there is an additional overestimation of MBO introduced by this paper.

I am willing to increase the score if the weaknesses and questions can be adequately addressed.

**Limitations:**

The limitations of the number of subtasks and how to split subtasks are adequately discussed in the paper.

---

> ### Author Rebuttal · Authors · 2023-08-08
>
> Thank you very much for the insightful comments.
>
> **(1) the terms "inner level" and "outer level", and the raised three questions.**
>
> Thank you for your valuable suggestion. We will include it in our revision. Thank you very much!
>
> **(2) Comparison to related works in offline RL with test time adaptation.**
>
> We thank the reviewer for raising the related work. We refer the reviewer to the global response for the comparison.
>
>
> **(3) In Figure 5, is it possible to also include the inference time of other baselines (COMs, RvS-R, etc)?**
>
> We believe that it is not very necessary to include the inference time of COMs or RvS-R because they only perform inference at s_0 (the initial state of a test rollout). The subsequent test time steps do not require any additional inference about the context variables. Therefore, we did not include their inference time in Figure 5.
>
> **(4) How do Onestep and DROP compare when they use the same amount of computation?**
>
> Thank you for this suggestion. We implement this comparison by searching for a time interval that makes DROP's inference time consistent with Onestep, and then we compare Onestep and DROP+CVAE (with the searched inference interval, DROP-Grad-Ada implementation). We show the experimental results in the following table (D4RL *-v2). We can see that overall, with the same amount of computation, DROP still performs better than Onestep, except for the tasks walker2d-medium-expert and walker2d-medium-replay.
>
> |     | Onestep  | DROP  |
> |  ----  |  ----  |  ----  |
> |  umaze  | 64.3  | **75.0**$\pm$2.3  |
> |  umaze-diverse  | 60.7  | **66.2**$\pm$1.9  |
> |  antmaze-medium-play  | 0.3  | **33.2**$\pm$3.6  |
> |  antmaze-medium-diverse  | 0  | **38.1**$\pm$2.5  |
> |  antmaze-large-play  | 0.3  | **21.5**$\pm$1.6  |
> |  antmaze-large-diverse  | 0  | **28.9**$\pm$3.1  |
> |  walker2d-medium-replay  | **66.4**  | 63.5$\pm$1.5  |
> |  hopper-medium-replay  | 77.3 | **78.8**$\pm$1.7 |
> |  halfcheetah-medium-replay  | 38.4  | **42.0**$\pm$1.9  |
> |  walker2d-medium-expert  | **111.8**  | 102.7$\pm$1.7  |
> |  hopper-medium-expert  | 81.4  | **100.2**$\pm$2.3  |
> |  halfcheetah-medium-expert  | 77.0  | **89.1**$\pm$1.8  |
>
>
>
> **(5) procedure of determining hyperparameters (the number of subtasks).**
>
> In our experiments, we take the number of subsets as a hyperparameter and choose it with typical hyperparameter tuning strategies. Thank you for pointing this out. We will clarify this in our revisions.
>
> We also point out that our DROP+CVAE treats each trajectory as a single subtask and can achieve better performance. Thus, in the real world, the algorithm designer does not need to worry much about the number of subtasks, just take each trajectory as a single subtask (and use the DROP+CVAE implementation).
>
>
> **(6) In Table 2, which variant is the listed DROP?**
>
> We use DROP-Grad-Ada in Table 2.
>
>
> **(7) When comparing DROP with LAPO in Table 2, why are the results of LAPO on u, um-p, and ul-p missing?**
>
> The main reason is that LAPO doesn't release their source code, and our private reproduction was not as good as the results in their paper (there may be some details we didn't notice). So we reported their results directly and then left it open if LAPO did not report their results in the paper.
>
> To provide a reference for the reviewer, we provide our reproduction results (antmaze *-v2) in the following table.
> |  |  umaze  |  umaze-diverse  |  medium-play  |  medium-diverse  |  large-play |  large-diverse  |
> |  ----  |  ----  |  ----  |  ----  |  ----  |  ----  |  ----  |
> LAPO (in its original paper, *-v1) |– |91.3 |– |**85.7** |– |61.7 |
> LAPO (our implementation, *-v2) | 86.5 $\pm$ 1.5 |80.6$\pm$2.1 |68.5$\pm$1.8 |79.2 $\pm$ 1.8| 48.8$\pm$ 2.6 |**64.8**$\pm$ 3.0 |
> DROP (*-v2) |**90.5**$\pm$2.4 |**92.2**$\pm$1.7 |**74.1**$\pm$3.9 |82.9$\pm$3.5 |**57.2**$\pm$5.5 |63.3$\pm$2.4 |
>
>
> **(8) results of Diffuser.**
>
> Thank you for pointing this out. We will include the Diffuser results in table 2 in our revision.
>
> **(9) why the iterative process can cause error propagation for potential overestimation of values**
>
> The main reason is that naively performing policy evaluation (Equation 1 in the main paper) may query the estimated Q^k(s', a') for actions that are far outside the static offline data, resulting in pathological value Q^{k+1}(s, a) with large error. Such an iterative process (iterative policy evaluation and improvement) will further cause the inferred policy \pi_{k+1}(a|s) to be biased towards OOD actions with erroneously overestimated values. Thus, performing policy evaluation and improvement iteratively leads to the (potential) policy/value errors being propagated iteratively, which in turn leads to collapsed performance.

---

> > ### Comment · Reviewer_SsTF · 2023-08-12
> >
> > Thanks for answering the questions and providing the new results. With the new results, I am now convinced by the advantage of DROP over OneStep with the same computation budget. I am also convinced by the novelty of the proposed algorithm and its possibility of introducing MBO as a general framework in offline RL. However, from the general response, it seems the the improvement of DROP over previous methods such as APE-V[1] and CCVL[2] are not very clear as acknowledged. It also seems that the method is a bit complicated with many hyperparameters that need to be tuned such as number of sub-tasks and the dimension of information bottleneck, so I'm not sure how we can tune those hyperparameters in real-life situations in the absence of a simulator. Therefore, I have decided to keep my original score of 5.
> >
> > [1] Ghosh, D., Ajay, A., Agrawal, P., & Levine, S. (2022, June). Offline rl policies should be trained to be adaptive. ICML 2022
> >
> > [2] Hong, J., Kumar, A., & Levine, S. (2022). Confidence-Conditioned Value Functions for Offline Reinforcement Learning. ICLR 2023

---

> > > ### Author Response · Authors · 2023-08-13
> > > **Thank you for the response.**
> > >
> > >
> > > We thank the reviewer for the response. We are glad to see that the reviewer acknowledges the novelty of the proposed algorithm and its possibility of introducing MBO as a general framework in offline RL. Thank you!
> > >
> > > **1. comparison to APE-V [1] and CCVL [2].**
> > >
> > > Regarding the relationship of DROP to APE-V [1] and CCVL [2], our approach in algorithm design is quite different from theirs: DROP is a non-iterative offline algorithm (which naturally avoids error propagation and exploitation), however, APE-V and CCVL use an iterative paradigm. Despite the fact that we both have the advantages of test-time adaptation, a clear benefit of our approach is that the OOD issues for offline RL are naturally solved in the non-iterative paradigm, while APE-V and CCVL require to be built on top of other offline RL methods to eliminate OOD problems, i.e., APE-V is based on Q-ensemble and CCVL utilizes the anti-exploration bonus.
> > > Therefore, in comparison to APE-V and CCVL, our algorithm is simpler and easier to implement.
> > >
> > > **2. application in real-life tasks.**
> > >
> > > We provide further clarification regarding the reviewer's concern about the implementation in real-life tasks.
> > >
> > > We note that the reviewer's main concern is the selection of the number of sub-tasks and the dimension of information bottleneck (dim(z)), and we acknowledge that our naive DROP implementation does have this worry. However, we must point out that `our DROP+CVAE implementation is totally innocent of such concerns`, and at the same time works better than naive DROP.
> > > + The number of sub-tasks: DROP+CVAE treats each trajectory in the offline data as a separate task, and thus one does not need to manually tune the number of sub-tasks.
> > > + The dimension of information bottleneck: we experimentally found that DROP+CVAE is in fact robust to dim(z), and a simple choice of 5 can lead to good performance on a large number of tasks. Therefore, the size of z is not worth being a concern for the practical application of DROP+CVAE.
> > >
> > > Therefore, in practice, one can just use the DROP+CVAE implementation, which completely eliminates the hyperparameter concerns mentioned by the reviewer and at the same time works better.
> > >
> > >
> > >
> > >
> > > We would like to thank the reviewer for the thoughtful comments. Please let us know if there are any concerns preventing you from raising your score.
> > >
> > > [1] Ghosh, D., Ajay, A., Agrawal, P., & Levine, S. (2022, June). Offline rl policies should be trained to be adaptive. ICML 2022
> > >
> > > [2] Hong, J., Kumar, A., & Levine, S. (2022). Confidence-Conditioned Value Functions for Offline Reinforcement Learning. ICLR 2023

---

### Official Review · Reviewer_hhjh · 2023-06-22

**Soundness:** 4 excellent
**Presentation:** 2 fair
**Contribution:** 4 excellent
**Rating:** 7
**Confidence:** 4

**Summary:**

The paper proposes a non-iterative offline RL algorithm, in which the policy is trained in a bi-level optimisation process. As opposed to commonly known iterative algorithms, the authors propose to split the optimisation into an inner loop training to mitigate any OOD issues, and an outer loop optimisation for reward maximisation. The paper compares the proposed algorithm on a subset of the D4RL benchmark datasets with prior iterative offline RL algorithms and demonstrates new state-of-the-art performance.

**Strengths:**

I think the proposed way of thinking about offline RL is still relatively new and a logical, very promising way forward. It seems wasteful to not make use of the information presented to the policy online (which is what many offline algorithms do since after training the policy stays fixed no matter what happens) - moving a part of the optimisation into this outer loop thus makes a lot of sense to me. A particularly new element to me is that the value that the final policy conditions on is not something handcrafted, but a learned representation of the potential behaviour policies. The combination with MBO is also a very interesting concept that I am unaware of being used in prior offline RL methods. I thus consider the method to score well in terms of novelty. I also believe that in the future we will see a lot more algorithms operating in a similar fashion, at least in the regard that part of the optimisation is moved to the online / outer loop.

The paper provides meaningful experiments and statistical analysis thereof: The algorithm is evaluated on maze as well as robotic locomotion tasks from the D4RL suite and the authors report measures beyond mean performance, such as probability of improvement and IQM performance. They also perform an analysis of the computational burden.

**Weaknesses:**

While I very much like the idea, I personally found the framing with the three questions and answers rather confusing. E.g. the questions are framed differently in abstract and introduction and leave room for lots of interpretation. For example, What should we pay attention to when exploiting the transferred information in the outer-level optimization? -> The question lacks a goal (i.e. what should we pay attention to ... in order to achieve WHAT?, otherwise the answer could be anything).

Also, the term "non-iterative, bi-level optimisation" is used very often. It is obvious what is meant by bi-level (inner and outer are described), but it is at least a little ambiguous what is meant by "non-iterative". I'm sure DROP also has iterations in some sense (i.e. mini batches during the learning of the conditional behaviour policy) - I think what you mean is that there is no iteratively learned value function which bootstraps off of itself and thus propagates errors further and further. Is that right? Please clarify.

The answer to question 3 is that the outer loop allows using the online data to adapt the policy. This is intuitive, but I would say an experiment to really show this is lacking. DROP is compared with some iterative baselines and shows better performance, but this could be due to other differences as well. What I would have liked to see would be something like:
- do everything like DROP, but try to put the outer-loop directly into the optimisation
- e.g.: perform optimisation in z-space for all states in the dataset and distill the resulting global policy model + conditioning values into a fixed model
- then you could isolate the difference that the outer-loop actually makes

The authors mention and qualitatively compare prior "non-iterative" algorithms (RvS, onestep, COMs, F-BC), yet do not include them in their empirical evaluation, which seems a little odd since one could assume they have in a way similar qualities and it should be interesting what benefit the newly proposed non-iterative method has over prior methods of this type.
Further, it seems to me that some prior works are missing in this context, i.e. there exist already offline RL algorithms that condition on something which can then adaptively change the policy behaviour by altering the value that it conditions on: In [1], the policy conditions on a level of confidence, in [2] on a balance between conservatism and more liberal behaviour, and in [3] it conditions on a belief over the MDP dynamics, which is updated based on each online state-action pair that is processed. I realise these are all rather recent, but I think they could at least be part of the qualitative comparison.

[1] Hong, J., Kumar, A., & Levine, S. (2022). Confidence-Conditioned Value Functions for Offline Reinforcement Learning. ICLR 2023

[2] Swazinna, P., Udluft, S., & Runkler, T. (2022). User-Interactive Offline Reinforcement Learning. ICLR 2023

[3] Ghosh, D., Ajay, A., Agrawal, P., & Levine, S. (2022, June). Offline rl policies should be trained to be adaptive. ICML 2022

Most weaknesses should be rather easy to address. What I am most concerned about is more relevant baselines (prior non-iterative algorithms, the missing ones that I mentioned, or the experiment proposed to actually show the merit of the outer-loop).

**Questions:**

How is the subset of D4RL datasets chosen? Why did you not use all of the available locomotion ones, e.g. medium-expert and medium-replay?

Why are there standard deviations only for DROP, but not for the other algorithms? How is the probability of improvement then calculated - is it whether the other methods mean is outside DROP's 95% CI or is it wither both 95% CI's don't overlap?

What do you mean exactly with non-iterative?

Why are the non-iterative prior works not part of the empirical evaluation?

**Limitations:**

Limitations are discussed & I enjoyed reading them since I also thought about these points while reading the paper.

potential negative societal impacts are not discussed

---

> ### Author Rebuttal · Authors · 2023-08-08
>
>
> Thank you very much for the insightful comments!
>
> **(1) description of the three questions and answers.**
>
> Thank you for the suggestion, we will clarify this in our revision of the paper.
>
> **(2) term: "non-iterative, bi-level optimization". I think what you mean is that there is no iteratively learned value function that bootstraps off of itself and thus propagates errors further and further.**
>
> Yes, you are right. We will clarify this in the introduction.
>
> **(3) comparison to a "distilled" DROP implementation, i.e., moving the outer-loop into the optimization and distilling the learned policy + conditions into a single fixed model.**
>
> Thank you for such a valuable suggestion! As you suggested, we have performed new experimental comparisons under the DROP+CVAE implementation (performing inference with DROP-Grad-Ada). We can see that such a "distilled" DROP implementation achieves competitive performance, approaching DROP's overall score (though still slightly below DROP). It really is a good proposal!
>
> |    | DROP  | "distilled" DROP   |
> |  ----  | ----  | ----  |
> |  walker2d-random-v2  | **5.2** $\pm$1.6  | 4.5 $\pm$0.8  |
> |  hopper-random-v2  | **20.8** $\pm$0.3  | 18.9 $\pm$0.2  |
> |  halfcheetah-random-v2  | **32.0** $\pm$2.5 | 27.8 $\pm$1.54  |
> |  walker2d-medium-v2  | 82.1 $\pm$5.2  | **84.2** $\pm$2.4  |
> |  hopper-medium-v2  | **74.9** $\pm$2.8  | 67.1 $\pm$2.1  |
> |  halfcheetah-medium-v2  | **52.4** $\pm$2.2  | 50.9 $\pm$1.7  |
> |  sum  | **267.4**  | 255.4  |
>
>
> **(4) The authors mention and qualitatively compare prior "non-iterative" algorithms (RvS, Onestep, COMs, F-BC), yet do not include them in their empirical evaluation.**
>
> We refer the reviewer to Figure 4, where we provide the empirical comparison to these baselines.
>
> **(5) comparison to recent works [1,2,3].**
>
> We thank the reviewer for raising these related works. We refer the reviewer to the global response for the comparison.
>
>
> [1] Confidence-Conditioned Value Functions for Offline Reinforcement Learning.
>
> [2] User-Interactive Offline Reinforcement Learning.
>
> [3] Offline rl policies should be trained to be adaptive.
>
> **(6) Why did you not use all of the available locomotion ones?**
>
> This is mainly due to page limitations, so we present only some of the comparative experiments in the main text, and include more results in the appendix.
>
> **(7) standard deviations for the other algorithms?**
>
> Thank you for pointing this out. The mean of most of the baselines is outside of DROP's 95% CI. We will include this in our revision. Thank you very much.
>
> **(8) What do you mean exactly with non-iterative?**
>
> We know that offline RL is prone to exploiting OOD state-actions and producing over-estimated values, which makes vanilla *iterative* policy/value optimization challenging. To eliminate the problem, a number of methods propose to 1) introduce a policy/value regularization in the iterative loop or 2) try to eliminate the iterative loop itself. The term “non-iterative” refers to the paradigm of eliminating the iteration loop.
>
> **(9) Why are the non-iterative prior works not part of the empirical evaluation?**
>
> The main starting point is that we have already compared DROP and non-iterative methods in Figure 4, so we did not list the non-iterative results in Table 2. Thank you for raising this point. We will include it in our revision.

---

> > ### Comment · Reviewer_hhjh · 2023-08-14
> > **Rebuttal Response**
> >
> > I thank the authors for their detailed response, which has helped to increase my understanding of the manuscript.
> >
> > A few questions remain:
> >
> > On Q3: If you say that the performance of the distilled DROP is almost as good as the full version, doesn't that mean the "outer loop during testing" isn't as important to performance as initially proposed in your paper? I see the distilled version is slightly worse, so a little advantage appears to persist, but not as much as one would have thought. How do you interpret these new results?
> >
> > On Q7: I realise you cannot pose revisions now, but could you provide the standard deviations in a table here?

---

> > > ### Author Response · Authors · 2023-08-17
> > >
> > > Thank you for the response!
> > >
> > > **1. Doesn't that mean the "outer loop during testing" isn't as important to performance as initially proposed in your paper?**
> > >
> > > It should not be seen that way. Actually, this "distilled" DROP is still an implementation of DROP: even if we move the policy improvement phase to the training phase, it is still a non-iterative offline RL method, and the outer-loop still exists. The main difference between the vanilla DROP and the "distilled" DROP is that: the vanilla DROP only conducts policy improvement on the samples encountered during the test (deployment phase) and then outputs the corresponding actions; the "distilled" DROP actually conducts policy improvement on all samples in the training data, and then distills the improved policies into a fixed policy. Therefore, there is still an outer-loop in the "distilled" DROP method, just that this outer-loop is moved to the training phase, not saying that the outer-loop is not important. The "distilled" DROP implementation is still the non-iterative offline RL method we advocated.
> > >
> > > **2. The standard deviations.**
> > >
> > > In the table below, we provide a performance comparison of the models on the Gym task. We also add new experimental results on the (Walker2d/Hopper/Halfcheetah) expert domains. (For a fair comparison, we take the baseline results from the LAPO paper.)
> > >
> > >
> > > |	|PLAS|LAPO|CQL|IQL|DROP|
> > > |---|---|---|---|---|---|
> > > |	Walker2d-random-v2	|	**9.2**	$\pm$	0.3	|	1.3	$\pm$	2.1	|	-0.2	$\pm$	0.1	|	5.4	$\pm$	0.4	|	5.2	$\pm$	1.6	|
> > > |	Hopper-random-v2	|	6.7	$\pm$	0	|	**23.5**	$\pm$	0.6	|	8.3	$\pm$	1.3	|	7.9	$\pm$	1.7	|	20.8	$\pm$	0.3	|
> > > |	Halfcheetah-random-v2	|	26.5	$\pm$	0	|	30.6	$\pm$	0.2	|	22.2	$\pm$	1.4	|	13.1	$\pm$	0.8	|	**32**	$\pm$	2.5	|
> > > |	Walker2d-medium-v2	|	75.5	$\pm$	10.6	|	80.8	$\pm$	0.8	|	**82.1**	$\pm$	6.3	|	77.9	$\pm$	2.5	|	**82.1**	$\pm$	5.2	|
> > > |	Hopper-medium-v2	|	51	$\pm$	4.3	|	51.6	$\pm$	3.3	|	71.6	$\pm$	10.3	|	65.8	$\pm$	5	|	**74.9**	$\pm$	2.8	|
> > > |	Halfcheetah-medium-v2	|	44.5	$\pm$	0.4	|	46	$\pm$	0.3	|	49.8	$\pm$	0.5	|	47.8	$\pm$	0.8	|	**52.4**	$\pm$	2.2	|
> > > |	Walker2d-expert-v2	|	109.6	$\pm$	0.6	|	**112.3**	$\pm$	0.1	|	108.8	$\pm$	0	|	110	$\pm$	0.3	|	110.8	$\pm$	2	|
> > > |	Hopper-expert-v2	|	107	$\pm$	11.6	|	106.8	$\pm$	3.7	|	102.3	$\pm$	7.5	|	109.4	$\pm$	2.3	|	**113**	$\pm$	3.9	|
> > > |	Halfcheetah-expert-v2	|	93.8	$\pm$	7.4	|	95.9	$\pm$	0.2	|	87.4	$\pm$	20.3	|	95	$\pm$	0.8	|	**99.7**	$\pm$	2.6	|
> > > |	sum	|	523.8			|	548.8			|	532.3			|	532.3			|	**590.9**			|
> > >
> > >
> > > ---
> > >
> > >
> > > We would like to thank the reviewer for the thoughtful comments. Please let us know if there are any concerns preventing you from raising your score.

---

> > > > ### Comment · Reviewer_hhjh · 2023-08-18
> > > > **Rebuttal Response 2**
> > > >
> > > > Thank you for your insights on the distilled DROP, the additional data, as well as addressing all my other concerns. I will increase my score to a 7.

---

### Official Review · Reviewer_7FJj · 2023-07-05

**Soundness:** 3 good
**Presentation:** 2 fair
**Contribution:** 3 good
**Rating:** 6
**Confidence:** 3

**Summary:**

The paper tackles offline reinforcement learning, and considers two classes of algorithms — iterative (at each step, policy evaluation and policy improvement is done sequentially) and non-iterative. The main contribution of the paper is to move the part of policy improvement from offline training to during test-time. This can increase the computational time of deploying the model, but seems to improve performance. The paper also make clever use of task decomposition and learning a low-dimensional task embedding to improve their results.

**Strengths:**

1. The idea is novel and very interesting.
2. The experimentations of the paper is thorough and organized, even though potentially some ablations can be added.
3. The paper is also written well, has high clarity and readability.


**Weaknesses:**

1. The final algorithm has many moving parts and components, and can feel to be over-engineered.
2. Ablation studies are not sufficient. One can potentially add ablations for effects of the hyper-parameters, and for example, it would be very important to see how the number of subtasks affect the performance of the algorithm.


**Questions:**

1. How is the task decomposition done for appendix section B.1? I.e., which of the three decomposition rules from B.2 is used? In general, more information about the task decomposition in the main paper would be appreciated.
2. How is the sequential sampling done for Rank(N, M) task decomposition? My guess is first M trajectories (with highest returns) become task 1, the next M trajectories become task 2 and so on.
3. Appendix line 048, “rank decomposition rule leverages more high quality trajectories” → what defines the quality of a trajectory? If quality = return of the trajectory, then wouldn’t some subtasks have lower quality in rank decomposition due to having low return trajectories only?
4. Could we have an ablation to see the effect of N on the performance of the algorithm? For example, the onestop algorithm [1] does not have a sub-task level decomposition. Maybe checking if there is a certain threshold for N, below which the onestop algorithm performs better, would be interesting. In other words, I am curious about how much improvement is coming from task divisions, if that is one of the main driving force for the given algorithm’s effectiveness.
5. Also is the computation time affected by N?
6. What is the advantage of using task embeddings $z$ instead of directly fitting a MBO on the collected data?
7. In table 2 (main paper), the umaze environment is antmaze-umaze or maze2d-umaze? Also the version of the environments should be included in the main paper.


[1] David Brandfonbrener, Will Whitney, Rajesh Ranganath, and Joan Bruna. Offline RL without off-policy evaluation. Advances in Neural Information Processing Systems, 34:4933–4946, 2021.

---

> ### Author Rebuttal · Authors · 2023-08-08
>
> Thank you very much for the positive comments.
>
> **Q1: which of the three decomposition rules from B.2 is used?**
>
> We use Rank(N, M) in the main paper. See Line 035 in the appendix. We will clarify it in our main paper. Thank you.
>
> **Q2: Rank(N, M) rule: My guess is first M trajectories (with the highest returns) become task 1, the next M trajectories become task 2, and so on.**
>
> Yes, you are right!
>
> **Q3: Appendix line 048, “rank decomposition rule leverages more high-quality trajectories” → what defines the quality of a trajectory? If quality = return of the trajectory, then wouldn’t some subtasks have lower quality in rank decomposition due to having low return trajectories only?**
>
> Yes, you are right. It is true that some subtasks may contain only low-return trajectories. The main motivation is that we expect to build *a diverse set of task distributions*. Then trajectories of different qualities will have different latent embeddings, which will benefit contextual policy learning and optimal embedding inference (test-time adaptation).
>
> **Q4: an ablation to see the effect of N on the performance.**
>
> Good suggestion. We report the ablation results in the following table.
> We can see that when N is very small, DROP basically performs worse than Onestep, proving that when the number of subtasks is small, updating z when doing outer-level optimization does not bring as much benefit as updating the policy directly (Onestep). A general trend can be observed that DROP becomes progressively better as N increases and outperforms Onestep for some values, suggesting that optimizing for low-dimensional z (out-level inference) brings more benefits at these points. However, we also note that larger N is not always better, and that performance degradation can occur when N is too large. We speculate that this is due to the fact that when N is too large, the learning (of the corresponding behavioral policies and Q-values) is underfitting and instead leads to worse performance in the end.
>
> |    | 1  | 2  | 5  | 8  | 10  | 20  | 50  | 100  | Onestep  |
> |  ----  | ----  | ----  | ----  | ----  | ----  | ----  | ----  | ----  | ----  |
> |  hopper-medium-replay-v2  | 65.9| 67.6  | 82.0| 79.5  | **87.4**  | 87.0  | 85.9  | 77.8  | 77.3 |
> |  halfcheetah-medium-replay-v2  | 33.8| 35.7| 34.8| 35.8| 34.1| 39.5  | **40.3**  | 38.6  | 38.3 |
> |  walker2d-medium-replay-v2  | 53.8| 57.9| 60.6| 59.7  | 61.9  | 59.9| 60.6| 56.9  | **66.4** |
>
>
>
> **Q5: Is the computation time affected by N?**
>
> In fact, N has a negligible effect on the inference time.
>
> **Q6: What is the advantage of using task embeddings instead of directly fitting an MBO on the collected data?**
>
> The main advantage of explicitly using task embedding is that we can perform test-time adaptation by exploiting the sequential structure of RL tasks, rather than simply performing inference at the beginning of the test rollout (which is what we do when we fit a simple MBO to the collected data). Empirically, we also find that test-time adaptation can yield better results than fitting a simple MBO (i.e., baseline COMs).
>
> **Q7: the version of the environments in Table 2.**
>
> All results are on the "v2" version of the datasets, except for the results of LAPO on antmaze, which are on the "v1" dataset. To unify the versions of the dataset used, we re-implement the LAPO algorithm (not released by the authors) and report the results. We refer the reviewer to question (7) in the authors' response to reviewer SsTF for our reproduced results. We will include the new results in our revision of the paper.

---

> > ### Comment · Reviewer_7FJj · 2023-08-11
> >
> > I thank the authors for carefully considering my concerns and trying to address them!
> >
> > **Further questions**:
> >
> > (1) **Related to Q4**: an ablation to see the effect of N on the performance.
> >
> > Would it be possible to repeat the experiment results for multiple seeds, and report an error bar? (Maybe for the purposes of the rebuttal, the authors do not need to test for so many values of N). It seems that for halfcheetah-medium-replay-v2, the values for N=20, 50,100 is very similar to Onestep, and without a proper error bar, bolding the N = 50 value to indicate superior result does not seem okay.
> >
> > (2) **Related to Q5**: Is the computation time affected by N?
> >
> > What is the computation time during training/learning the task embeddings with respect to N? In general, between Onestep [1] and DROP on similar tasks, how would the training time scale?
> >
> > (3) **Related to Q3**: Appendix line 048, “rank decomposition rule leverages more high-quality trajectories” → what defines the quality of a trajectory? If quality = return of the trajectory, then wouldn’t some subtasks have lower quality in rank decomposition due to having low return trajectories only?
> >
> > Thanks to the authors for their answer. However, this still does not clarify why line 048 says “rank decomposition rule leverages more high-quality trajectories” if some subtasks have lower quality in rank decomposition. What do the authors mean when they say "... leverages more high-quality trajectories". A comparison of the average/standard deviation of subtask trajectory quality between the three rules would clarify this issue. I imagine that might be out-of-scope for this paper, in which case changing this statement appropriately would be important.
> >
> > (4) **Appendix C: Best hyperparameters**
> >
> > I thank the authors for reporting the hyper-param search grid in the appendix. However, in addition, would it be possible to report the best hyper-params for each environment? This would help reproduce the paper's results efficiently.
> >
> > (5) **Appendix C: Cost of hyper-param tuning**
> >
> > One of the strong points of Onestep [1] was its robustness to hyper-params. Could the authors provide a brief discussion on all the hyper-params that Onestep needs to tune, and all the hyper-params DROP needs to tune? It seems, from Appendix C, that on D4RL environments, a default choice of hyper-params do not work for all environments. The hyper-param tuning strategy mentioned also seems pretty extensive, with a two step strategy. A more thorough investigation on the robustness of DROP to hyper-parameter choice would be appreciated.
> >
> > (6) **Appendix C: Step 2 of hyper-param tuning**
> >
> > Is having access to a simulator for hyper-param tuning during offline training a valid assumption?
> >
> > (7) **Additional question 1**: Have the authors considered using something other than CVAE to model offline data and multiple-modes? For example, [2] discusses transformers to model multiple modes in offline data. Any reason for choosing CVAEs in particular?
> >
> >
> > [1] Offline RL Without Off-Policy Evaluation, https://arxiv.org/abs/2106.08909
> >
> > [2] Behavior Transformers: Cloning modes with one stone, https://proceedings.neurips.cc/paper_files/paper/2022/file/90d17e882adbdda42349db6f50123817-Paper-Conference.pdf

---

> > > ### Author Response · Authors · 2023-08-13
> > > **Thank you for the response**
> > >
> > >
> > > We thank the reviewer for the response. Below we address your further questions one by one.
> > >
> > > **(1) Would it be possible to repeat the experiment results for multiple seeds, and report an error bar?**
> > >
> > > Thank you for the suggestion. We provide the results (with 4 seeds) in the following table.
> > >
> > > | | 1 | 2 | 5 | 8 | 10 | 20 | 50 | 100 | Onestep |
> > > | ---- | ---- | ---- | ---- | ---- | ---- | ---- | ---- | ---- | ---- |
> > > | hopper-medium-replay-v2 |64.1	$\pm$2.4	|68.7	$\pm$1.4	|82.6 $\pm$2.1	|79.7	$\pm$1.2	|**86.6**	$\pm$1.2	|85.1	$\pm$0.8	|84.9	$\pm$1.1 	|78.3	$\pm$2.3| 77.3 |
> > > | halfcheetah-medium-replay-v2 |35.7	$\pm$1.3	|34.8	$\pm$0.5	|35.4	$\pm$1.5	|36.0 $\pm$1.2	|36.5	$\pm$2.3	|39.2	$\pm$2.1	|**42.5**	$\pm$2.5	|39.1	$\pm$1.3| 38.3 |
> > > | walker2d-medium-replay-v2 |52.1	$\pm$2.1	|56.0	$\pm$0.8	|61.3	$\pm$1.4	|58.7	$\pm$2.1	|63.1	$\pm$1.8	|60.5	$\pm$2.9	|60.2	$\pm$2.7	|57.2	$\pm$2.0| **66.4** |
> > >
> > >
> > > **(2) Is the computation time affected by N?**
> > >
> > > At inference (test-time adaptation), we record the average computation time of DROP when varying the size of sub-tasks (N). We can observe that N has a negligible effect on the inference time (since we can do parallel computations).
> > >
> > > | average inference time (s) during testing  | N=10 | N=50 | N=100 | N=200 | N=500 | N=1000 |
> > > | ---- | ---- | ---- | ---- | ---- | ---- | ---- |
> > > | time interval = 10| 0.017| 0.0171| 0.0171| 0.0171| 0.0173| 0.0174|
> > > | time interval = 20| 0.0108| 0.0109| 0.0109| 0.011| 0.0111| 0.0112|
> > > | time interval = 50| 0.0088| 0.0087| 0.0087| 0.0088| 0.0091| 0.0091|
> > > | time interval = 100| 0.0073| 0.0075| 0.0075| 0.0076| 0.0078| 0.008|
> > >
> > >
> > >
> > > **(3) “rank decomposition rule leverages more high-quality trajectories.”**
> > >
> > > Thank you for pointing out this. Indeed, such a statement is not very rigorous, and we will revise it in our revision. Thank you!
> > >
> > > **(4) Would it be possible to report the best hyper-parameters for each environment?**
> > >
> > > Thank you. We refer the reviewer to Table 5 (in the appendix) for the best hyper-parameters for each environment. We will clarify it in our revision.
> > >
> > > **(5) A more thorough investigation on the hyper-parameter choice would be appreciated.**
> > >
> > > Thank you for the suggestion. Compared to other offline RL methods, it is true that our naive DROP implementation introduces an extra hyper-parameter, the number of subtasks, due to the task decomposition step. However, we point out that our DROP+CVAE does not need to additionally tune this hyper-parameter at all. This is because DROP+CVAE treats each trajectory in the offline data as a separate task. Meanwhile, DROP+CVAE is also robust to dim(z) (the dimension of information bottleneck), and a simple choice of 5 can lead to good performance on a large number of tasks. Compared to other offline RL algorithms, training DROP+CVAE does not introduce any additional burden on hyperparameter selection.
> > >
> > > Therefore, DROP+CVAE implementation can eliminate the additional worry regarding the hyper-parameters, and it works better compared to naive DROP. In real-life offline RL tasks, one can just use the DROP+CVAE implementation.
> > >
> > >
> > > **(6) Is having access to a simulator for hyper-parameter tuning during offline training a valid assumption?**
> > >
> > > Yes. Such a setting for hyper-parameter tuning is consistent with most offline RL papers.
> > >
> > >
> > > **(7) Any reason for choosing CVAEs in particular?**
> > >
> > > The main reason for such a choice is the ability of CVAE models to yield compact and low-dimensional embeddings (robust and widely adopted), making them suitable for deriving test-time adaptation in our non-iterative paradigm.
> > > Paper [1] mentioned by the reviewer does focus on multi-modal data, which however assumes such data is expert. Therefore, there is no explicit policy improvement, which is the fundamental difference between [1] and DROP. Of course, it is absolutely feasible to replace the MLP policy architecture with a transformer model. Thanks to the review for suggesting this, we will discuss it further in our revision.
> > >
> > >
> > > [1] Shafiullah, N. M., Cui, Z., Altanzaya, A. A., & Pinto, L. Behavior Transformers: Cloning k modes with one stone. NeurIPS 2022.
> > >
> > >
> > > We would like to thank the reviewer for the thoughtful comments. Please let us know if there are any concerns preventing you from raising your score.

---

> > > > ### Comment · Reviewer_7FJj · 2023-08-13
> > > >
> > > > I thank the reviewers for finishing experiments in such a short amount of time!
> > > >
> > > > I have the following comments/questions:
> > > >
> > > > 1. **Is the computation time affected by N?**
> > > >
> > > > I think the authors misunderstood my question. I was interested in training, not test-time adaptation. For example, on a certain offline training dataset, how much time would the Onestep algorithm take to converge vs DROP? How does that scale with N? To clarify, I am interested in **DROP (training)** instead of **DROP (testing/deployment)** in algorithm 3.5.
> > > >
> > > > 2. **Environment**
> > > >
> > > > For the sake of completeness, which environment is used for the computation time vs N experiment reported above?
> > > >
> > > > 3. **Choice of best hyper-params**
> > > >
> > > > I thank the authors for including these in the appendix, and apologize for missing them in my earlier reading!
> > > >
> > > > 4. **Having access to a simulator for hyper-parameter tuning**
> > > >
> > > > Upon further investigation, it does seem consistent with most offline RL paper. However, I would suggest adding a few prior work as citation and justify this assumption in the paper to make it stronger.

---

> > > > > ### Author Response · Authors · 2023-08-13
> > > > > **We thank the reviewer for the quick response.**
> > > > >
> > > > > We thank the reviewer for the quick response.
> > > > >
> > > > > **(1) Is the computation time affected by N?**
> > > > >
> > > > > Thank you for the clarification. We would like to shortly state our opinion on the reviewer's concern: it seems that offline algorithms often don't care about the training time, and only online RL algorithms care about that (they need to compare the sample efficiency). That's why basically most offline RL papers mainly provide tables to show the final performance, and then online RL papers provide results with figures, which aims to compare the performance of different models under the same training time (sample efficiency). Therefore, we respectfully believe that it makes little sense to compare training times (not RL large models) for offline RL algorithms. Meanwhile, in this test-time adaptation scenario, the inference computation time is more important.
> > > > >
> > > > > Yet, we provide the reviewer here with the approximate training time of our algorithm --- for different numbers of N, the training time for a single task (D4RL gym domains) is around 8 hours (Tesla-V100-SXM2-32GB). We apologize for not recording checkpoints of the model during training, and therefore cannot immediately provide an accurate training time when convergence (it might be shorter than 10 hours).
> > > > >
> > > > > In summary, we will fully incorporate the review's suggestions and provide such comparisons in our revision. We sincerely appreciate your suggestions for improving this paper. Thank you!
> > > > >
> > > > > **(2) Which environment is used for the computation time vs N experiment reported above?**
> > > > >
> > > > > We use halfcheetah-random.  The environment has a minimal effect on the inference (testing) time because different environments simply differ in the dimensions of the input states and the output actions.
> > > > >
> > > > > **(3 \& 4) hyper-parameter.**
> > > > >
> > > > > Thank you for the suggestion.  We will clarify it in our revision.

---

> > > > > > ### Comment · Reviewer_7FJj · 2023-08-13
> > > > > >
> > > > > > I understand the point about training time not being important for offline algorithms --- I was merely curious about the effect if N is different/higher.
> > > > > >
> > > > > > Thanks to the authors for taking the time to address all my concerns!

---

> > > > > > > ### Author Response · Authors · 2023-08-13
> > > > > > > **Glad to see that we have addressed all your concerns**
> > > > > > >
> > > > > > > Thanks again for reviewing our submission and we are glad to see that we have addressed your concerns. According to our early training results, the impact of N on training time is relatively small (considering that our policy training is essentially a simple supervised regression). Across a large range of N, DROP can generally achieve competitive results within a maximum of 8 hours.
> > > > > > >
> > > > > > > Thanks again for all of your constructive suggestions, which have helped us improve the quality and clarity of our paper.

---

### Official Review · Reviewer_YuS3 · 2023-07-06

**Soundness:** 2 fair
**Presentation:** 2 fair
**Contribution:** 2 fair
**Rating:** 3
**Confidence:** 4

**Summary:**

This paper proposes a method, namely DROP, to adapt the policy during the inference time. The authors achieve this by dividing the optimization process into two phases. In the first phase, the authors train a contextual behavior policy, a score model, and a deterministic task embedding model. At the second phase, the authors utilzie the score model for adapting the policy execution, i.e., follow the optimal embedding. The authors belives their proposed method can better achieve "stitching" across the states in the offline data. They provide some experiments on D4RL datasets to show the benefits of their proposed method.

**Strengths:**

(a) this paper is well-structured and easy to follow

(b) the figures are nice and helpful for the readers to understand the claims from the authors

(c) it is good to see the authors consider statistical uncertainty and significance in the paper. I believe this is quite important to RL fields


**Weaknesses:**

(a) I have several concerns on the ideas presented in the main text

 - the adopted method shares many similarities with CQL, i.e., the authors utilize a CQL-style optimization objective in Equation 8. The regularization term is hence not novel. The differences are, CQL penalizes actions, while DROP penalizes the embedding $z$.

 - though the author claim that they learn a *score model*, the learnt score function $f$ is actually ***action-value function***, but conditioned on the task embedding. This can be observed in Eqaution 8, where the score model is updated via bellman error. This is highly similar to [1, 2]. Both [1] learns a value function conditioned on the confidence $\delta$, while [2] learns the value function conditioned on the evolving belief. These two papers are relevant to this paper, while they are not cited in this paper. I believe they are important baselines to compare against. Specially, [1,2] can also achieve the claimed *test-time adaption* by adjusting the confidence or the belief. Actually, this has already been achieved by [2]. To realize this in [1], perhaps another objective function is needed (or simply by maximizing the confidence conditioned value function).

 - the authors cite RoMA [3], but do not involve it as a baseline. I can tell that RoMA is also strongly related to the topics covered in this paper, and RoMa seems to exhibit better performance than COMs

 - the generated embeddings seem to serve as *goal* for the learnt policy. I believe some goal-conditioned and return-conditioned algorithms ought to be included **numerically** as baselines. The authors compare their DROP against IQL, CQL, PLAS, LAPO while none of them are goal-conditioned or return-conditioned algorithms. The authors provide IQM comparison against decision transformer, RvS-R, while more advanced and stronger methods are expected

 - Meanwhile, the comparison in Table 1 is misleading. These criterias are manually proposed by the authors. The success of DROP in answering A1-A3 do not necessiarily indicate that DROP is better than prior methods

 - the proposed method, DROP, seems quite redundant. It requires to first split the offline dataset into some subsets, and then learn the embedding network and the contextual behavior policy upon it. Additionally, one needs to learn a score model, and finally utilizes this score model for querying the optimal embedding during inference. It is unclear whether there is a need for dividing the datasets into some subsets and learn embeddings on them. The authors provide a design choice comparison concerning on which decomposition rule is better, while the corresponding analysis on whether decomposition itself is needed is missing. The authors write that "suggests that fitting a single behavior policy may not be optimal to model the multiple modes of the offline data distribution" (line 125-126), but the authors themselves choose to learn one single contextual behavior policy instead of multiple policies conditioned on the task embedding. Since the behavior policy is unique, why do we still need to split the datasets? For example, why not the authors directly receive $s,a$ as inputs to train $\phi$. I believe this eliminates the need of dividing the datasets. Also, the reviewer does not think that training loss is a good indicator to show that multiple policies can better characterize the offline datasets (as presented in appendix B.1).

[1] Confidence-conditioned value functions for offline reinforcement learning

[2] Offline rl policies should be trained to be adaptive

[3] Roma: Robust model adaptation for offline model-based optimization

(b) no theoretical understandings are provided either in the main text or in the appendix. I do not want to blame the authors too much on this point. However, it is unclear whether the socre function will overestimate even with the introduced regularization, especially during deployment (i.e., whether poor embedding is provided even if we optimize the score function at inference time). It is also unclear how large should we penalize the embedding. One of the advantages of CQL is that it theoretically answers that one ought to use a large $\alpha$ to enforce conservatism. The authors comment that the inference can be unstable if the number of subtasks is large. I believe the instability is due to the wrong output embedding. It will be better if the authors can theoretically characterize the relationship between the embedding and the number of subsets.

(c) I also have the following concerns on the experiment section

 - in Table 2 of main text, some baseline results are lower than their typically reported ones in prior work, e.g., CQL

 - in Table 2 of main text, the results of LAPO on antmaze are on "v1" version datasets, and I can tell that the results of IQL on antmaze are from "v2" datasets. It is unclear whether the results of baselines and DROP are also conducted on "v2" datasets, since the authors do not specify the version of datasets in the main text. This is huge problematic.

 - in Table 2 of main text, it is unclear which variant of DROP is used to report the results, given the fact that the authors introduce four variants of DROP in section 5

 - I can see that the authors use MuJoCo "v0" datasets in Figure 2 while "v2" in Table 1 in the appendix. You should unify the versions of dataset used.

 - in line 72, the authors write that "achieves better performance compared to state-of-the-art offline RL algorithms". However, the compared baselines are NOT state-of-the-arts, especially on MuJoCo datasets. The authors ought to use this term carefully.

 - the superior performance of DROP are acquired via a very careful tuning of hyperparameters. Based on Table 5 in the appendix, the authors carefully search for the optimal number of subsets $N$ for each dataset. This value varies across different datasets, and may lies in a large range (e.g., 1000 subsets on halfcheetah-medium-expert, but 50 subsets on halfcheetah-medium-replay). The choice of the number of subsets is quite weird. For example, the halfcheetah-medium-expert dataset only contains two modes, while DROP requires 1000 subsets to learn a contextual behavior policy. Meanwhile, it is unclear how this value affects the performance of DROP practically. The authors say that "we find that when the number of sub-tasks is too large, the inference is unstable", but how unstable is it? I believe this is a quite important hyperparamter, and its parameter study ought to be included. The need of searching the best number of subsets decreases the practicality of DROP

 - based on the table 3 in the appendix, the authors use a large batch size 1024 for DROP. As far as the reviewer can tell, the baseline algorithms all use a batch size of 256. A recent work show that a larger batch size is helpful for offline RL learning [4], hence I believe the comparison of DROP against other methods is not fair

 - it is difficult for the user to find the best time interval when deploying DROP in practice, and the best values may differ across different scenarios

[4] Q-Ensemble for Offline RL: Don't Scale the Ensemble, Scale the Batch Size

(d) It is unclear whether the proposed method can be applied in the real-world applications, where we may require a fast response from the policy. Since DROP requries test-time adaption, I think at the current stage it is not suitable to be deployed in the real-world robots. Moreover, in real-world scenarios, we actually expect the robots to be able to adapt their policies during deployment, as it is highly possible that the robots encounter the unseen scenarios (e.g., different landforms) and need to adapt the policies. I do not see the potential of using DROP to mitigate these challenges. The generalization capability of DROP to unseen scenarios is limited.

(e) Minor issues

 - the format of appendix seems to be another venue instead of NeurIPS

 - appendix A contains nothing

**Questions:**

(a) this paper does not seem to involve any theoretical analysis, why do you choose `yes` to `theory`?

(b) it seems the performance of DROP can be largely affected by the performance of the behavior policy, can it be improved if we train the contextual behavior policy in a IQL-style manner, i.e., weighted behavior cloning

**Limitations:**

The authors discuss several limitations of their work, and the reviewer personally agrees with that.

---

> ### Author Rebuttal · Authors · 2023-08-08
>
> **About Weaknesses:**
>
> **(a): concerns about the ideas.**
> + *the difference between CQL and DROP:* The main difference is that DROP performs non-iterative bi-level offline RL optimization, while CQL performs iterative bi-level offline RL optimization (see Figure 1 in the main paper).
> + *comparison to [1, 2, 3] and more advanced algorithms:*  Please refer to the global response for the comparison.
> + *goal-conditioned methods ought to be included numerically:* Thank you for the suggestion. We will include it in our revision.
> + *stronger methods are expected:* See CCVL in the  global response.
> + *criteria in Table 1:* First, in Appendix B.5, we have conducted an ablation study to examine the impact of the conservative regularization used to learn the score model (Q2A2), and find that removing such regularization leads to unstable performance, thus demonstrating the effectiveness of Q2A2. Second, in Figure 3, we also demonstrate the effectiveness of answering Q3A3, i.e. that performing adaptive policy inference can produce better results.
> + *concerns on the task decomposition rule:* First, we point out that the purpose of performing task decomposition is to establish a link between offline RL and MBO, thereby analyzing different offline RL methods from a unified perspective. Second, the purpose of performing task decomposition (using returns) is to provide a fair comparison with previous approaches (i.e., RvS) in the non-iterative paradigm, thereby demonstrating the superiority of DROP's design (by answering Q1Q2Q3). Third, DROP can treat each trajectory as a single task, as in our DROP+CVAE implementation (lines 296-306), which does not require complex task decomposition. Our experiments also show the effectiveness of DROP. Thus, the algorithm designer does not need to worry much about the choice of decomposition rule in deployment, just take the DROP+CVAE implementation (treating each trajectory as a single subtask).
>
>
>
>
> **(b) whether the score function will overestimate even with the introduced regularization**
>
> If we assume that the problem (overestimation, raised by the verification) occurs, there is actually an implicit advantage of DROP: the overestimation occurs on top of the inference to z, not on top of the actions. That is, even if the inference yields an OOD z, the contextual behaviour policy still has the ability to produce in-distribution actions. We have also done an ablation study on this in Appendix D (lines 383-428) and find that the learned behaviour policy can protect against the shift of the inferred embeddings. Similarly, the same benefit (implicit OOD regularisation) is also observed in the PLAS paper. Thus, we believe that there is no need to be overly concerned about DROP's overestimation.
>
>
> **(c) concerns on the experiment section.**
> + *CQL results:* Our CQL results in Antmaze domain are taken from the RvS paper and results in Gym domain are taken from the LAPO paper.
> + *Environment version in Table 2:* All results are on the "v2" version datasets, except the results of LAPO on antmaze, which are on the "v1" dataset. To unify the dataset versions used, we re-implement the LAPO algorithm (not released by the authors) and report the results. We refer the reviewer to question (7) in the authors' response to reviewer SsTF for our reproduced results. We will include them in our revision of the paper.
> + *which variant of DROP in Table 2:* DROP-Grad-Ada.
> + *unify the versions of the dataset used:* Thank you for the suggestion. We will unify it in our paper revision.
> + *searching the best number of subsets:* In our experiments, we take the number of subsets as a hyperparameter and choose it with typical hyperparameter tuning strategies. In response to the review's concern about algorithmic complexity, I don't think it's worth worrying about. Because in the implementation we can treat each trajectory as a separate task, similar to the DROP+CVAE implementation in the paper. This simple implementation can give better performance, and at the same time, the algorithm designer does not have to worry much about the decomposition rule.
> + *large batch size:* We respectfully point out that the issue raised by the review in paper [4] does not make sense. Paper [4] does emphasize the importance of large batch size, but this paper is still based on Q-ensemble (eliminates OOD problems). Our approach to OOD problems has nothing to do with this paper (and Q-ensemble ideas). Large batch size can improve the robustness of model training, and we respectfully don't think we need to force all offline RL work (and even future works) to set 256 sizes for the sake of "fairness".
>
> **(d) concerns on real-world applications.**
>
> We note that the reviewer's main concern is that real-world deployment requires real-time inference and that this is time consuming. We think the reviewer is overly concerned about this. First, even if our z is fixed (updated only at s_0), we have found experimentally that DROP still outperforms the RvS and decision-transform baselines. Second, we do not actually need to run inference every time. We found that we can still get good results by running inference every 10 steps. We tested the inference time on a very early RTX 2080 Ti, which only takes about 0.01 seconds to infer. In particular, in many of today's robotic arm tasks, the speed of the arm's movement is quite slow, so 0.01 seconds will basically have no effect. Thirdly, our model only uses the MLP network, and the inference speed is much slower than that of transformer/diffusion based methods. Therefore, we do not believe that inference time will be an obstacle for DROP in real applications.
>
>
>
> **About Questions:**
>
> **(a) Why do you choose yes to theory?**
>
> We apologize for any confusion about this. We will correct it.
>
> **(b) train the contextual behavior policy in an IQL-style manner**
>
> We thank the reviewer for pointing out this option. We refer the reviewer to the global response for the comparison.

---

> > ### Comment · Reviewer_YuS3 · 2023-08-15
> > **Official Comment by Reviewer YuS3**
> >
> > Thanks for your rebuttal. After carefully reading the paper (and appendix) again, reading the comments from other reviewers and the corresponding rebuttal from the authors, I decided to keep my current rating of 3. Please find the comments and suggestions below
> >
> > **On the method**
> >
> > - **The regularization term is not novel**. The authors utilize a CQL-style optimization objective. Though the authors reply that *DROP performs non-iterative while CQL performs iterative bi-level offline optimization*. Their differences are minor
> >
> > - **The learnt score function is action-value function $Q$ in nature**, but additionally conditioned on the task embedding
> >
> > - **Stronger goal-conditioned and return-conditioned methods ought to be included as baselines**, e.g., Diffuser and its variants
> >
> > - **On the task decomposition rule**. The authors reply that *DROP can treat each trajectory as a single task, which does not require complex task decomposition*. This raises the following concerns: (a) the sampled starting points can be large, making the test time adaptation of DROP quite slow (b) the authors write in line 346 that **when the number of sub-tasks is too large, the inference is unstable**. It is unclear how unstable it is and whether DROP+CVAE can *absolutely* address this issue (c) many trajectories have similar returns (they should have similar task embedding), and $z$ can be diverse if the dataset is large
> >
> > - **DROP introduces too many hyperparameters**. I still believe DROP is redundant and introduces too many critical hyperparameters, e.g., number of subtasks, inference time intervals, learning rate in Eqn 9, Lagrange threshold, dimension of embedding, etc. Finding a balance between these parameters is hard when the simulator is absent
> >
> > - **Potential overestimation in score function**. The authors reply that *even if the inference yields an OOD z, the contextual behavior policy still has the ability to produce in-distribution actions*. It is understandable since the contextual policy is trained via behavior cloning. However, that does not indicate that the potential overestimation does not count in DROP. In Appendix D Figure 6, the performance of DROP drops drastically on halfcheetah-medium and hopper-medium without regularization term. That being said, **the overestimation in score function can significantly affect the performance of DROP**. It is natural since the task embedding $z$ is output by querying the score function, and wrong embedding can be fed into the contextual policy if the score function overestimates. Also, one can see in Table 6 in the appendix that DROP fails in relocate-expert dataset and has a poor performance on hammer-expert dataset. I think the reasons can also be attributed to the overestimation in score function. These altogether make the practicality of DROP doubtable and limited.
> >
> > **On the clarity of the paper**
> >
> > - the authors ought to unify the versions of the dataset used and specify what variant of DROP they use to prevent any misunderstanding. MuJoCo -v0 datasets can be removed as they have bugs. DROP has too many variants and some of them can be discussed in the appendix
> >
> > - the authors should run CQL with d3rlpy implementation then
> >
> > - given the potential benefits of large batch size, it is better to use a batch size of 256
> >
> > - no theoretical analysis is included. From a pure engineering perspective, it is okay that one searches for the best number of subsets, or simply treat each trajectory as a subtask, while from a Neurips paper of this kind, I expect to understand why DROP works with a chosen number of subtasks. Theoretical analysis can make this paper stronger
> >
> > - the speed can be an obstacle in real applications if we treat each trajectory as a subtask and the dataset is large
> >
> > - the authors show in Appendix B.1 that multiple behavior policies deliver better resilience to characterize the offline data than a single behavior policy. However, they train a single contextual policy in DROP and this gap ought to be further clarified
> >
> > **Suggestions**
> >
> > - include the comparison and discussions against APE-V, CCVL, RoMA in the later version
> >
> > - include a numerical comparison against some goal-conditioned and return-conditioned algorithms
> >
> > - I understand that the authors want to show the advantages of DROP over prior methods, while answering Q1 Q2 Q3 (e.g., the authors write in line 190 that *However, both F-BC and RvS-R leave Q2 unanswered*) do not necessarily indicate that prior methods are flawed
> >
> > - LAPO has an official codebase, the authors ought to use that instead of your implementation
> >
> > - parameter study on the number of subtasks is recommended
> >
> > - IQL-style manner means that you can update the contextual policy via $\mathbb{E}[\exp(k A(s,a,z))\log\beta(s|a,z)]$, $k$ is the inverse temperature, $A$ is the advantage. To that end, you need to train an extra value function $V(s,z)$ (no regularization on score function is needed then)
> >
> > I hope that the authors can find some of them helpful

---

> > > ### Author Response · Authors · 2023-08-17
> > >
> > > We thank the reviewer for the detailed suggestion. I would like to make the following clarifications.
> > >
> > > **1. regularization term.**
> > >
> > > We did not emphasize that this is our contribution. We just used it in our implementation. Our main contribution (novelty) is to propose a non-iterative offline RL paradigm from the perspective of MBO (as pointed out by Reviewers 7FJj, hhjh, and SsTF).
> > >
> > >
> > > **2. The learned score function is an action-value function $Q$ in nature, additionally conditioned on the task embedding.**
> > >
> > > Yes, you are right.
> > >
> > > **3. It is unclear how unstable it is and whether DROP+CVAE can absolutely address this issue (unstable) many trajectories have similar returns (they should have similar task embedding), and $z$ can be diverse if the dataset is large.**
> > >
> > > The source of instability is that the naive DROP implementation additionally introduces a one-hot coding process, which leads to unstable learning if the number (dimension of one-hot) is too large. However, DROP+CVAE does not introduce any additional one-hot encodings and learns the task embeddings directly through CVAE, hence `its learning is stable`. Also, since CVAE explicitly introduces a Gaussian prior for bounding the learned embeddings, `there will be no such a problem as mentioned by the reviewer` that if the data is too large $z$ will be very diverse.
> > >
> > > **4. Hyper-parameters on 1) the number of subtasks, 2) inference time intervals, 3) the learning rate in Equation 9, 4) the Lagrange threshold, and 5) the dimension of embedding.**
> > >
> > > + 1). DROP+CVAE does not need to manually select the parameter of the number of subtasks.
> > > + 2). An inference time interval setting of 10 can achieve a balance between model performance and inference time on most tasks at the same time.
> > > + 3). The learning rate in Equation 9 does not need to be set separately, and it is fine to keep it consistent with the model training learning rate.
> > > + 4). The Lagrange threshold is a hyper-parameter. We choose it with the standard hyper-parameters selection rules. Empirically, we did not observe a significant difference in performance across a range of values. Therefore, we simply set it as 2 (see details in Implementation Details in the appendix). Therefore, our model is robust against this parameter.
> > > + 5). In experiments, we didn't spend much time tuning the dimension of $z$ (CVAE’s learning is stable), and a simple setting of 5 on most tasks is sufficient.
> > >
> > > In summary, all these hyper-parameters mentioned by the reviewer `are not bottlenecks affecting the deployment` of the algorithm in real-life tasks.
> > >
> > > **5. Potential overestimation in score function.**
> > >
> > > We would like to emphasize that there is a potential overestimation on some tasks, so we introduced regularization (over the score function) to eliminate this overestimation problem. `Besides, through ablation studies, we also found that this regularization does eliminate potential overestimation`. At the same time, we would like to point out that we did not emphasize that it (the regularization) is our contribution. Regularization is useful for eliminating overestimation in offline RL, and thus we introduced it in our method design.
> > >
> > >
> > > **6. Regarding clarity and suggestions.**
> > >
> > > Thank you very much for your suggestions to improve the quality of this paper. We sincerely accept all your suggestions. At the same time, we note that the current rebuttal/discussion phase of NeurIPS does not support modifying submissions, and we will incorporate all your suggestions into our revision. Meanwhile, we note that the modification does not involve the core contribution of our paper, and we hope you are satisfied with the main contribution of this work.
> > >
> > >
> > > ---
> > >
> > > We would like to thank the reviewer for the thoughtful comments. Please let us know if there are any concerns preventing you from raising your score.

---

> > > > ### Comment · Reviewer_YuS3 · 2023-08-20
> > > >
> > > > Thanks for the additional response, though my opinion is fairly unchanged. This paper has severe flaws in **clarity** (experiments, statement), comparison with prior related work, and deeper analysis and insights
> > > >
> > > > > We did not emphasize that regularization term is our contribution
> > > >
> > > > Cannot agree with that, the regularization term is clearly a critical component of DROP and is a key contribution
> > > >
> > > > > On the task decomposition rule
> > > >
> > > > The authors respond that DROP+CVAE can mitigate many concerns like *manually selecting the parameter of the number of subtasks*, and *potential overestimation in score function*. Then **I do not see any reason for discussing the task decomposition rule and listing it as the main component in DROP** (simply treating each trajectory and learning task embedding via CVAE can be good and the discussed limitation can be addressed). The initial motivation and the suggested DROP+CVAE implementation have a large gap. The motivation of this paper is, fitting a single behavior policy may not be optimal to model the multiple modes of the offline data distribution (Section 3.1 and Appendix B), hence task decomposition is needed to exploit the hybrid modes in offline data. However, unless I have overlooked something, DROP+CVAE fits a **single** policy and treats each trajectory as a subtask. Moreover, in DROP+CVAE, **it is unclear what the task embedding represents**. In the vanilla DROP, we know that $z$ is the embedding of a specified task, while if we learn $z$ by using Eqn 1 in Appendix B.4, it is unclear what it represents now. The authors commented that *CVAE explicitly introduces a Gaussian prior for bounding the learned embeddings*. It is somewhat confusing as the prior distribution of task embedding is not necessarily Gaussian, and it is questionable that the learned $\phi(z)$ can always distinguish different tasks and guarantee that the policy can be improved during deployment. Finally, since DROP+CVAE is not claimed as the main algorithm and contribution, my concerns about vanilla DROP still remain (e.g., instability).
> > > >
> > > > > DROP introduces too many hyperparameters
> > > >
> > > > I do not doubt that in the simple simulated tasks presented in this paper, DROP can achieve good performance with some fixed hyperparameters. However, things may not be true if we use DROP on more complex, real-world tasks. Introducing too many hyperparameters itself limits the practicality of DROP. As commented, theoretical justifications of DROP can make the claims from the authors stronger
> > > >
> > > > > Baselines and related work comparison
> > > >
> > > > DROP shares many similarities with APE-V, CCVL. DROP realizes test-time adaptation via outer-loop optimization (finding the task embedding), while APE-V adapts its policy through belief. Detailed discussion and comparison are needed to show the advantages of DROP. It is clear that DROP is compatible with ensemble
> > > >
> > > > Baselines can be added, e.g., RoMA, stronger goal-conditioned and return-conditioned methods
> > > >
> > > > > Potential overestimation
> > > >
> > > > I still think the potential overestimation can occur, even with the regularization term. The failure of DROP on relocate-expert and hammer-expert datasets validates that. Hence, I still doubt the practicality of DROP. As an additional note, from Figure 6 in the appendix, it is clear that DROP-Grad-Ada achieves an average score lower than 100 on walker2d-medium-expert, while in Table 6, you report 106.9 on it. Maybe you should check all of your results in the main text and appendix
> > > >
> > > > > On the clarity of the paper
> > > >
> > > > Clarity is a huge drawback of this work. As commented, the experiments are often misleading as the versions of the dataset are not unified or even different in Table 2, the baseline results (CQL) seem to be lower, and the batch size is not aligned. Meanwhile, it is often unclear where the authors use DROP+CVAE implementation, and which variant of DROP is used for comparison (there are many variants of DROP)
> > > >
> > > > > Deeper analysis of DROP
> > > >
> > > > The authors claim that DROP+CVAE is a better choice for implementation, while the corresponding analysis on DROP+CVAE is missing. The motivation and rationality of treating each trajectory as a single task and learning a single policy upon the task embeddings ought to be specified. I can tell from the paper and the authors' response that, okay DROP can improve test-time performance by querying the task embedding, but how can the authors guarantee that the policy can always be improved? I think no such guarantee can be claimed. Meanwhile, it seems clear that the performance of DROP is largely affected by the learned behavior policy. Perhaps, the authors can analyze how different behavior policies affect the performance of DROP
> > > >
> > > > > On test-time adaptation
> > > >
> > > > As commented in the initial review, what I expect for test-time adaptation is the ability to adapt to unseen scenarios. However, I do not see such potential in DROP. If under the same environment, it is unclear whether DROP can be a better choice over offline2online RL algorithms

---

> > > > > ### Author Response · Authors · 2023-08-20
> > > > > **[Clarification 1/2] Thank you for improving this paper.**
> > > > >
> > > > > We thank the reviewer for the detailed suggestion. I would like to make the following clarifications.
> > > > >
> > > > > **1. The regularization term is clearly a critical component of DROP and is a key contribution**
> > > > >
> > > > > We agree that conservative regularization is a critical component of DROP, but we did not emphasize that it is not the main contribution of this paper. `The main contribution of our paper is to propose and unify the non-iterative bi-level offline RL optimization paradigm, from the perspective of model-based optimization (MBO).` Same as in this non-iterative paradigm, our main contribution is compared to RvS-R, Onestep, Decision Transformer, and COMs. In Table 1, we compare them from the methodological perspective, and in Figure 4, we compare them quantitatively.
> > > > >
> > > > >
> > > > > **2. The reason for discussing the task decomposition rule and listing it as the main component in DROP.**
> > > > >
> > > > > The reason is that we want to formulate offline RL as an offline model-based optimization (MBO) paradigm.
> > > > >
> > > > > **3. The motivation of this paper is, fitting a single behavior policy may not be optimal to model the multiple modes of the offline data distribution, hence task decomposition is needed to exploit the hybrid modes in offline data.**
> > > > >
> > > > >
> > > > > `This is a huge misunderstanding.` That's not our motivation. Our motivation is to explore other alternatives to offline RL from the view of MBO (model-based optimization). Modeling the multiple modes is just an additional benefit when we bring offline RL into offline MBO. Please see Line 123, we are describing "additionally" and "also comes with the benefit", which means that it is just an additional benefit under the MBO formulation, not the main motivation of this paper.
> > > > >
> > > > >
> > > > > **4. DROP+CVAE fits a single policy and treats each trajectory as a subtask.**
> > > > >
> > > > >
> > > > > We kindly point out that DROP+CVAE fits a single **contextual** policy.
> > > > >
> > > > >
> > > > > **5. In DROP+CVAE, what does the task embedding represent?**
> > > > >
> > > > > The task embeddings represent some statistics of the trajectory. For example, Decision Transformer [1] simply takes the return as the statistics of the trajectory, while our DROP uses the CVAE to learn such embeddings (statistics). This is consistent with the paper [2].
> > > > >
> > > > >
> > > > > [1] Chen, Lili, et al. "Decision transformer: Reinforcement learning via sequence modeling." NeurIPS 2021.
> > > > >
> > > > > [2] Furuta, Hiroki, Yutaka Matsuo, and Shixiang Shane Gu. "Generalized decision transformer for offline hindsight information matching." ICLR 2022.
> > > > >
> > > > >
> > > > > **6. It is questionable that the learned $\phi(z)$ can always distinguish different tasks**
> > > > >
> > > > > There is also reconstruction loss inside CVAE, which will explicitly encourage $z$ to capture task-relevant embeddings (which will be used for decoding). As long as the tasks (trajectories) are different, the learned z can be well distinguished.
> > > > >
> > > > >
> > > > > **7. DROP+CVAE is not claimed as the main algorithm and contribution.**
> > > > >
> > > > > We thank the reviewers for recognizing that our DROP+CVAE can solve all the above issues. The main reason is that we would like to introduce the MBO paradigm to offlineRL while unifying previous non-iterative offline RL methods. If we directly describe DROP+CVAE in the paper, this is not friendly to readers who are not familiar with MBO. The logic of our paper is to first introduce MBO, then introduce vanilla DROP, then introduce the implementation of DROP+CVAE, gradually expanding the implementation of MBO in offline RL and eliminating potential issues. Meanwhile, in practice, one can just use the DROP+CVAE implementation, which eliminates the instability concerns mentioned by the reviewer and at the same time works better.
> > > > >
> > > > >
> > > > > **8. Hyperparameters.**
> > > > >
> > > > >
> > > > > In practice, one can just use the DROP+CVAE implementation, which `eliminates the hyperparameter concerns` mentioned by the reviewer and at the same time works better.
> > > > >
> > > > >
> > > > > **9. Baselines and related work comparison.**
> > > > >
> > > > > Thank you for the suggestion. In our revision, we will add new discussions to APE-V, CCVL, and RoMA (see our rebuttal comparison results in the global response).
> > > > >
> > > > >
> > > > > **10. Results in Figure 6 and Table 6 appendix.**
> > > > >
> > > > > Thank you for the careful checking! We'll check it carefully. Thank you!

---

> > > > > ### Author Response · Authors · 2023-08-20
> > > > > **[Clarification 2/2] Thank you for improving this paper.**
> > > > >
> > > > > **11. The versions of the dataset.**
> > > > >
> > > > > For the results in the main text, we have uniformly used the v2 version of the dataset (we regret that we can’t change the paper right now, due to the rules of NeurIPS). In the table below, we provide a performance comparison of the models on the Gym task. We also add new experimental results on the (Walker2d/Hopper/Halfcheetah) expert domains. (For a fair comparison, we take the baseline results from the LAPO paper.) We hope the reviewer will be satisfied with this.
> > > > >
> > > > >
> > > > > | |PLAS|LAPO|CQL|IQL|DROP|
> > > > > |---|---|---|---|---|---|
> > > > > | Walker2d-random-v2 | **9.2** $\pm$ 0.3 | 1.3 $\pm$ 2.1 | -0.2 $\pm$ 0.1 | 5.4 $\pm$ 0.4 | 5.2 $\pm$ 1.6 |
> > > > > | Hopper-random-v2 | 6.7 $\pm$ 0 | **23.5** $\pm$ 0.6 | 8.3 $\pm$ 1.3 | 7.9 $\pm$ 1.7 | 20.8 $\pm$ 0.3 |
> > > > > | Halfcheetah-random-v2 | 26.5 $\pm$ 0 | 30.6 $\pm$ 0.2 | 22.2 $\pm$ 1.4 | 13.1 $\pm$ 0.8 | **32** $\pm$ 2.5 |
> > > > > | Walker2d-medium-v2 | 75.5 $\pm$ 10.6 | 80.8 $\pm$ 0.8 | **82.1** $\pm$ 6.3 | 77.9 $\pm$ 2.5 | **82.1** $\pm$ 5.2 |
> > > > > | Hopper-medium-v2 | 51 $\pm$ 4.3 | 51.6 $\pm$ 3.3 | 71.6 $\pm$ 10.3 | 65.8 $\pm$ 5 | **74.9** $\pm$ 2.8 |
> > > > > | Halfcheetah-medium-v2 | 44.5 $\pm$ 0.4 | 46 $\pm$ 0.3 | 49.8 $\pm$ 0.5 | 47.8 $\pm$ 0.8 | **52.4** $\pm$ 2.2 |
> > > > > | Walker2d-expert-v2 | 109.6 $\pm$ 0.6 | **112.3** $\pm$ 0.1 | 108.8 $\pm$ 0 | 110 $\pm$ 0.3 | 110.8 $\pm$ 2 |
> > > > > | Hopper-expert-v2 | 107 $\pm$ 11.6 | 106.8 $\pm$ 3.7 | 102.3 $\pm$ 7.5 | 109.4 $\pm$ 2.3 | **113** $\pm$ 3.9 |
> > > > > | Halfcheetah-expert-v2 | 93.8 $\pm$ 7.4 | 95.9 $\pm$ 0.2 | 87.4 $\pm$ 20.3 | 95 $\pm$ 0.8 | **99.7** $\pm$ 2.6 |
> > > > > | sum | 523.8 | 548.8 | 532.3 | 532.3 | **590.9** |
> > > > >
> > > > >
> > > > > **12. Where do the authors use DROP+CVAE implementation, and which variant of DROP is used for comparison?**
> > > > >
> > > > > In Table 2 (main text), we used the DROP+CVAE Grad-Ada implementation. We will clarify it in our revision.
> > > > >
> > > > >
> > > > >
> > > > > **13. The motivation and rationality of treating each trajectory as a single task.**
> > > > >
> > > > >
> > > > > Such treatment is consistent with the idea of hindsight information matching [1, 2], which aims to learn a contextual policy whose trajectory rollouts satisfy some desired information statistics (hindsight information).
> > > > >
> > > > >
> > > > > [1] Furuta, Hiroki, Yutaka Matsuo, and Shixiang Shane Gu. "Generalized decision transformer for offline hindsight information matching." ICLR 2022.
> > > > >
> > > > >
> > > > > **14. “Okay DROP can improve test-time performance by querying the task embedding, but how can the authors guarantee that the policy can always be improved?”**
> > > > >
> > > > > We clarify that such a policy improvement is implicitly satisfied by performing gradient ascent to find the optimal behavior embedding z* using the learned score function (contextual Q function). **Such treatment is consistent with Onestep**, with the only difference being that we query for a better embedding, however, Onestep optimizes the output actions directly.
> > > > >
> > > > > **15. The authors can analyze how different behavior policies affect performance.**
> > > > >
> > > > > We clarify that different task classification rules and different numbers of subtasks can produce different behavioral policies (as suggested by the Reviewer). Indeed different behavioral policies affect the final performance (see the influence of decomposition rules in Appendix B.1 and the number of subtasks in the rebuttal to Reviewer 7FJj). But it doesn't matter, the simplest DROP+CVAE, also producing a behavior policy, can achieve better results without introducing extra parameters.
> > > > >
> > > > >
> > > > > **16.The ability to adapt to unseen scenarios.**
> > > > >
> > > > > Thank you for your suggestion. To test the potential ability to adapt to unseen scenarios, we change the transition dynamics of the Gym environment (changing the height and mass of the agent's torso, respectively) and then compare the performance of DROP with Onestep, COMs, and RoMA in the new environment. We can see that our method achieves better results overall when deployed in the new environment compared to other baselines.
> > > > >
> > > > >
> > > > > | | Onestep | COMs | RoMA | DROP+CVAE (Grad-Ada) |
> > > > > |---|---|---|---|---|
> > > > > | Halfcheetah-medium-v2 (changing height) | 37.6 $\pm$ 2.2 | 27.6 $\pm$ 1.6 | 35.6 $\pm$ 2.1 | **48.1** $\pm$ 2.8|
> > > > > | Hopper-medium-v2 (changing height) | 64.0 $\pm$ 1.8 | 43.9 $\pm$ 2.2 | 39.0 $\pm$ 0.7 | **70.1** $\pm$ 1.6 |
> > > > > | Walker2d-medium-v2 (changing height) | 71.8 $\pm$ 0.7 | 28.5 $\pm$ 1.9 | 55.2 $\pm$ 2.9 | **75.8** $\pm$ 2.0 |
> > > > > | Halfcheetah-medium-v2 (changing mass) | **46.4** $\pm$ 2.8	|	20.1 $\pm$ 1.3	|	35.5 $\pm$ 3.6	|	44.3 $\pm$ 3.9	|
> > > > > | Hopper-medium-v2 (changing mass) | 54.7 $\pm$ 3.5	|	46.5 $\pm$ 1.8	|	53.7 $\pm$1.6	|	**60.5**	 $\pm$ 2.0|
> > > > > | Walker2d-medium-v2 (changing mass) | 66.2 $\pm$ 1.5	|	29.5 $\pm$ 2.4	|	58.5 $\pm$ 3.2	|	**79.3** $\pm$ 1.7	|
> > > > >
> > > > > ---
> > > > >
> > > > > We appreciate the reviewer's suggestions very much. We hope the above response (comparison with prior related work, and deeper analysis and insights) has addressed your concerns about the clarity of our paper. Thank you again for the thoughtful comments. Please let us know if there are any concerns preventing you from raising your score. Thank you!

---

> > > > > > ### Comment · Reviewer_YuS3 · 2023-08-21
> > > > > >
> > > > > > Thanks for the responses and your active engagement in the discussion phase. I think my rating is already.
> > > > > >
> > > > > > **Final notes**: it is quite clear that the clarity of this paper is under the bar of acceptance. The authors include many new experiments during the discussion phase, and these new results definitely make the paper stronger (I still have issues with the experiments though), but I wish these experiments were included in the original submission. It is like a new paper to me at the moment. The authors often reply to my concerns on DROP with the DROP+CVAE implementation, but it does not compensate for the inherent flaw in the vanilla DROP method. It is quite clear that DROP+CVAE is initially introduced for comparison with latent policy methods (lines 296-316) in the main text, and it does not even appear in the method part. The authors say that their logical flow is to first introduce MBO, then vanilla DROP, and then DROP+CVAE. However, it is clear that the role of DROP+CVAE is weak in the main text and is not properly and thoroughly discussed. My concerns about the task embedding in DROP+CVAE retains. As this paper does not involve theoretical analysis, I'd consider this paper an empirical paper, and correctness and clarity are vital then. Unfortunately, the clarity is less satisfying and there are bugs in the reported results. I don't want to underscore what the authors did and I certainly appreciate that, but it is quite hard to judge the paper now. Even if I increase my score, I'd still recommend this paper to be rejected as I don't see it is ready for prime time.
> > > > > >
> > > > > > Further comments on your responses can be found below
> > > > > >
> > > > > > > contribution
> > > > > >
> > > > > > The claimed contribution is somewhat not novel enough, APE-V does almost the same thing (outer-level optimization on policies by querying the value function)
> > > > > >
> > > > > > > reason for discussing the task decomposition rule
> > > > > >
> > > > > > I do not think that much space is needed to spare on discussing the task decomposition rule if you claim that DROP+CVAE is the ultimate solution
> > > > > >
> > > > > > > motivation of this paper
> > > > > >
> > > > > > I do not agree that there is a misunderstanding. To develop your method from the view of MBO, you propose to decompose the tasks and this is needed since fitting a single behavior policy may not be optimal
> > > > > >
> > > > > > > DROP+CVAE
> > > > > >
> > > > > > Though the authors claim that *the task embeddings represent some statistics of the trajectory*, the corresponding deeper discussions are not presented in the paper. Obviously, analysis on DROP+CVAE is absent (e.g., how the task embeddings affect the policy and $Q$). Meanwhile, a deep connection between DROP and [x] ought to be specified. More justifications are needed to show that the learned z can be well distinguished. It is also vital to see how robust $Q$ is concerning z
> > > > > >
> > > > > > [x] Generalized decision transformer for offline hindsight information matching
> > > > > >
> > > > > > > The motivation and rationality of treating each trajectory as a single task
> > > > > >
> > > > > > It is somewhat strange that you explain it as *the idea of hindsight information matching* but do not actively include detailed explanations in the paper. This is a critical point from my perspective
> > > > > >
> > > > > > > how can the authors guarantee that the policy can always be improved
> > > > > >
> > > > > > As commented, I believe no such guarantee can be made. The failure of DROP on relocate-expert and hammer-expert datasets validates that
> > > > > >
> > > > > > > The authors can analyze how different behavior policies affect performance
> > > > > >
> > > > > > The reviewer does target DROP+CVAE here. The authors can think about training the contextual policy using other methods other than CVAE, e.g., taking minimum over ensemble critics, using IQL-style policy, etc.
> > > > > >
> > > > > > > The ability to adapt to unseen scenarios
> > > > > >
> > > > > > I appreciate your new experiments on generalization to unseen scenarios. Let us take a look at the new experiments, the authors write that they changed the height and mass of the agent's torso, while it is unclear what are the detailed differences (e.g., what are the detailed changes to the height and mass?) Based on the results, the baseline algorithms can already achieve good performance. The advantages of DROP+CVAE over Onestep are limited on many tasks. This makes the reviewer doubt that the advantages of DROP+CVAE will be even smaller if stronger baselines are compared.
> > > > > >
> > > > > > The authors can also consider evaluating the DROP+CVAE upon unseen data (i.e., set the initial state to an unseen state), and evaluate its ability in adapting its policy.

---

> > > > > > > ### Author Response · Authors · 2023-08-21
> > > > > > > **Thank you for improving this paper**
> > > > > > >
> > > > > > > Dear reviewer YuS3,
> > > > > > >
> > > > > > > We sincerely thank the time and effort you have engaged in the review/discussion phase. We will incorporate all of your suggestions (new results and discussions) into our new version. Thanks again for your insightful comments and suggestions!
> > > > > > >
> > > > > > > Best,
> > > > > > >
> > > > > > > Authors

---

### Author Rebuttal · Authors · 2023-08-09




**(1) comparison to prior adaptive baselines.**

1) The following table shows a comparison with APE-V [1] in the D4RL suite. The APE-V results are taken from the original paper, which uses SAC-N as the base offline RL implementation. Despite the comprehensive approach we have taken, it is important to acknowledge that the performance of our DROP does not generally outperform the specific baseline APE-V [1]. Nevertheless, we believe that this result provides valuable insights for future research directions on MBO solutions for offline RL problems.

|        | APE-V [1]  | DROP |
|  ----  | ----  | ----  |
|  halfcheetah-random  | 29.9 $\pm$ 1.1  | 32.0 $\pm$ 2.5  |
|  halfcheetah-medium | 69.1 $\pm$ 0.4 | 52.4 $\pm$ 2.2  |
|  halfcheetah-medium-expert  | 101.4 $\pm$ 1.4  | 102.2 $\pm$ 1.5  |
|  halfcheetah-medium-replay  | 64.6 $\pm$ 0.9  | 50.9 $\pm$ 1.6  |
|  hopper-random  | 31.3 $\pm$ 0.2 | 20.8 $\pm$ 0.3  |
|  hopper-medium |   | 61.5 $\pm$ 3.7  |
|  hopper-medium-expert  | 105.7 $\pm$ 3.7 | 107.2 $\pm$ 1.5  |
|  hopper-medium-replay  | 98.5 $\pm$ 0.5 | 96.3 $\pm$ 2.5  |
|  walker2d-random  | 15.5 $\pm$ 8.5 | 5.2 $\pm$ 1.6  |
|  walker2d-medium | 90.3 $\pm$ 1.6 | 82.1 $\pm$ 5.2  |
|  walker2d-medium-expert  | 110.0 $\pm$ 1.5  | 109.3 $\pm$ 0.4  |
|  walker2d-medium-replay  | 82.9 $\pm$ 0.4 | 83.5 $\pm$ 1.2  |

2) As CCVL [2] only experiments on the discrete action Atari games and does not test its performance on the common continuous control tasks of D4RL. For comparison, we also deploy our DROP method on 4 offline Atari games. We present the results in the following table (using an initial 10\% of the replay dataset after 12.5M gradient steps). We can see that our DROP outperforms CCVL in 2 out of 4 tasks.

|          | CCVL [2]  | DROP  |
|  ----  | ----  | ----  |
|  Asterix | 7576.0 $\pm$ 360.2  | 6517.9 $\pm$ 564.4  |
|  Breakout  | 121.4 $\pm$ 10.3  | 139.5 $\pm$ 25.0  |
|  Pong  | 13.4 $\pm$ 6.1  | 10.7 $\pm$ 8.2  |
|  Seaquest  | 1211.4 $\pm$ 437.2  | 1358.1 $\pm$ 352.0  |



3) Regarding the paper [3] mentioned by the reviewer, we think it is difficult to make a fair experimental comparison. The main reason is that paper [3] asks the user to adjust the policy behaviour after training, which is too subjective and therefore we cannot make a fair comparison. We will cite this paper in our paper and explain the connection between us. Thank you again for bringing up this valuable related work.



[1] Ghosh, D., Ajay, A., Agrawal, P., & Levine, S. (2022, June). Offline rl policies should be trained to be adaptive. ICML 2022

[2] Hong, J., Kumar, A., & Levine, S. (2022). Confidence-Conditioned Value Functions for Offline Reinforcement Learning. ICLR 2023

[3] Swazinna, P., Udluft, S., & Runkler, T. (2022). User-Interactive Offline Reinforcement Learning. ICLR 2023



**(2) comparison to RoMA.**

As suggested by reviewer YuS3, we also make a comparison with RoMA [4]. In the implementation, we take the parameters (neural network weights) of the behavioural policies as the design input and the behavioural scores as the outputs. In the pre-training phase, we also use Gaussian noise for input perturbation; in the inference phase, we perform parameter inference (outer-level optimisation) with 200-step updates. The comparison results are shown in the following table. We can see that RoMA improves the performance of COMs and can outperform our DROP in 1 out of 3 tasks.


|        | COMs  | RoMA | DROP |
|  ----  | ----  | ----  | ----  |
|  halfcheetah-medium-replay  | 41.4  | 56.8 $\pm$ 2.9  | 50.9 $\pm$ 1.6  |
|  hopper-medium-replay  | 49.7| 61.2 $\pm$ 2.0  | 96.3 $\pm$ 2.5  |
|  walker2d-medium-replay  | 33.9 | 67.4 $\pm$ 4.1  | 83.5 $\pm$ 1.2  |


[4] Yu, S., Ahn, S., Song, L., & Shin, J. (2021). Roma: Robust model adaptation for offline model-based optimization. NeurIPS 2021.



**(3) train the contextual behavior policy in an IQL-style manner.**

We note that IQL essentially weights the behavior policy with Q-values, and also note that its policy is not a contextual policy, but rather a simple policy that inputs states and outputs actions. In this paradigm, we believe that a reasonable way to understand the review's proposal (IQL style) is to first perform a policy improvement for each contextual policy, and then distill multiple improved contextual policies into a single fixed policy. In the implementation, we first use the score function to optimize z (implicit policy improvement), and then distill the improved contextual policy into a fixed policy, in the same spirit as the Q-weighted BC in IQL. The following table shows the comparison results. We can see that this method ("distilled" DROP) is able to match DROP on walker2d-medium-v2 and halfcheetah-medium-v2, but it still performs worse than our non-iterative DROP implementation overall.


|    | DROP  | "distilled" DROP   |
|  ----  | ----  | ----  |
|  walker2d-random-v2  | **5.2** $\pm$1.6  | 4.5 $\pm$0.8  |
|  hopper-random-v2  | **20.8** $\pm$0.3  | 18.9 $\pm$0.2  |
|  halfcheetah-random-v2  | **32.0** $\pm$2.5 | 27.8 $\pm$1.54  |
|  walker2d-medium-v2  | 82.1 $\pm$5.2  | **84.2** $\pm$2.4  |
|  hopper-medium-v2  | **74.9** $\pm$2.8  | 67.1 $\pm$2.1  |
|  halfcheetah-medium-v2  | 52.4 $\pm$2.2  | **53.9** $\pm$1.7  |
|  sum  | **267.4**  | 258.4  |

---

### Decision · Program_Chairs · 2023-09-21

**Decision:**

Accept (poster)

**Comment:**

The paper introduces an approach designed for adapting policies during inference in offline reinforcement learning. It introduces a bi-level optimization process, which enhances policy adaptability by segregating inner and outer loop optimizations. Reviewers commend the paper for its well-structured presentation and insightful figures. The approach is recognized for its potential in transferring part of the optimization process to test-time. Furthermore, the paper's claims are substantiated by experiments conducted on D4RL benchmarks, which take into account both performance metrics and computational overhead.

Several concerns regarding hyperparameters and comparisons were raised. It is also identified that there are some inconsistency between the results shown in figures and tables.

Overall, the meta-reviewer recommends acceptance, contingent on that the authors carefully addressing the raised concerns and integrating reviewers' feedback..